

**1**  **Volatility and lifetime against OH heterogeneous reaction of ambient Isoprene Epoxydiols-**
**2**  **Derived Secondary Organic Aerosol (IEPOX-SOA)**

**3**  Weiwei Hu[1,2], Brett B. Palm[1,2], Douglas A. Day[1,2], Pedro Campuzano-Jost[1,2], Jordan E.
**4**  Krechmer[1,2], Zhe Peng[1,2], Suzane S. de Sá[3], Scot T. Martin[3,4], M. Lizabeth Alexander[5], Karsten
**5**  Baumann[6], Lina Hacker[7], Astrid Kiendler-Scharr[7], Abigail R. Koss[1,2,8], Joost A. de Gouw[1,2,8],
**6**  Allen H. Goldstein[9,10], Roger Seco[11], Steven J. Sjostedt[8], Jeong-Hoo Park[12], Alex B. Guenther[11],
**7**  Saewung Kim[11], Francesco Canonaco[13], André. S. H. Prévôt[13], William H. Brune[14], Jose L.
**8**  Jimenez[1,2]

**9**  1 Cooperative Institute for Research in Environmental Sciences, University of Colorado,
**10**  Boulder, CO, USA 80309
**11**  2 Department of Chemistry and Biochemistry, University of Colorado, Boulder, CO, USA 80309
**12**  3 John A. Paulson School of Engineering and Applied Sciences Harvard University, Cambridge,
**13**  MA, USA 01742
**14**  4 Department of Earth and Planetary Sciences, Harvard University, Cambridge, MA, USA 01742
**15**  5 Environmental Molecular Sciences Laboratory, Pacific Northwest National Laboratory,
**16**  Richland, WA, USA 99352
**17**  6 Atmospheric Research and Analysis Inc., Morrisville, NC, USA 27560
**18**  7 Institute for Energy and Climate Research - Troposphere (IEK-8), Forschungszentrum Jülich,
**19**  D-52425 Jülich, German
**20**  8 Earth System Research Laboratory, NOAA, Boulder, Colorado, USA 80305
**21**  9 Department of Environmental Science, Policy, and Management, University of California,
**22**  Berkeley, CA, USA 94720
**23**  10 Department of Civil and Environmental Engineering, University of California, Berkeley, CA,
**24**  USA 94720
**25**  11 Department of Earth System Science, University of California, Irvine, USA 92697
**26**  12 National Institute of Environmental Research, Republic of Korea 22689
**27**  13 Laboratory of Atmospheric Chemistry, Paul Scherrer Institute (PSI), 5232 Villigen,
**28**  Switzerland
**29**  14 Department of Meteorology, Pennsylvania State University, University Park, PA, USA 16802
**30**  Correspondence to: J. L. Jimenez (jose.jimenez@colorado.edu)

**31**  **Abstract**

**32**  Isoprene epoxydiols-derived secondary organic aerosol (IEPOX-SOA) can contribute
**33**  substantially to organic aerosol (OA) concentrations in forested areas under low NO conditions,
**34**  hence significantly influencing the regional and global OA budgets, accounting for example for
**35**  16-36% of the submicron OA in the SE US summer. Particle evaporation measurements from a
**36**  thermodenuder show that the volatility of ambient IEPOX-SOA is lower than that of bulk OA
**37**  and also much lower than that of known monomer IEPOX-SOA tracer species, indicating that
**38**  IEPOX-SOA likely exists mostly as oligomers in the aerosol phase. The OH aging process of
**39**  ambient IEPOX-SOA was investigated with an oxidation flow reactor (OFR). New IEPOX-SOA
**40**  formation in the reactor was negligible, as the OFR cannot accelerate processes such as aerosol
**41**  uptake and reactions that do not scale with OH. Simulation results indicate that adding ~100 μg
**42**  m$^{-3}$ of pure $H_2SO_4$ to the ambient air allows to efficiently form IEPOX-SOA in the reactor. The



heterogeneous reaction rate coefficient of ambient IEPOX-SOA with OH radical ($k_{OH}$) was
estimated as $4.0 \pm 2.0 \times 10^{-13}$ cm$^3$ molec$^{-1}$ s$^{-1}$, which is equivalent to more than a 2-week lifetime.
A similar $k_{OH}$ was found for measurements of OH oxidation of ambient Amazon forest air in an
OFR. At higher OH exposures in the reactor ($>1 \times 10^{12}$ molec. cm$^{-3}$ s), the mass loss of IEPOX-
SOA due to heterogeneous reaction was mainly due to revolatilization of fragmented reaction
products. We report for the first time OH reactive uptake coefficients ($\gamma_{OH}=0.59\pm0.33$ in SE US
and $\gamma_{OH}=0.68\pm0.38$ in Amazon) for SOA under ambient conditions. A relative humidity
dependence of $k_{OH}$ and $\gamma_{OH}$ was observed, consistent with surface area-limited OH uptake. No
decrease of $k_{OH}$ was observed as OH concentrations increased. These observation of
physicochemical properties of IEPOX-SOA can help to constrain OA impact on air quality and
climate.



## 1 Introduction

Organic aerosol (OA), which comprises 10-90% of ambient submicron aerosol mass globally, has important impacts on climate forcing and human health (Kanakidou et al., 2005; Zhang et al., 2007; Hallquist et al., 2009). However, quantitative predictions of OA mass concentrations often fails to match the real ambient measurements by large factors, (e.g.Volkamer et al., 2006; Dzepina et al., 2011; Tsigaridis et al., 2014). Improved characterization of the properties and lifetime of OA is needed to better constrain OA model predictions.

Isoprene is the most abundant non-methane hydrocarbon (NMHC) emitted into the Earth's atmosphere (Guenther et al., 2012). Many studies in the past decade have shown that the reaction products of isoprene-derived epoxydiols (IEPOX), formed under low NO conditions (Paulot et al., 2009), can contribute efficiently to secondary OA (SOA) via reactive uptake of gas-phase IEPOX onto acidic aerosols (Eddingsaas et al., 2010; Froyd et al., 2010; Surratt et al., 2010; Lin et al., 2012; Liao et al., 2015). IEPOX-SOA measurements in field studies show that it can account for 6-34% of total OA over multiple forested areas across the globe, with important impacts on the global and regional OA budget (Hu et al., 2015). Although the formation of IEPOX-SOA from gas-phase IEPOX has been investigated in many laboratory studies (e.g. Eddingsaas et al., 2010; Lin et al., 2012; Gaston et al., 2014), the lifetime and aging of IEPOX-SOA in the aerosol phase is still mostly unexplored in the literature.

IEPOX-SOA can be measured by multiple methods. Gas chromatography/mass spectrometry (GC/MS) or liquid chromatography/mass spectrometry (LC/MS) of filter extracts can be used to measure some IEPOX-SOA species (accounting for 8-80% of total IEPOX-SOA depending on the study, Lin et al., 2012; Budisulistiorini et al., 2015; Hu et al., 2015). Recently, several studies have shown that factor analysis of real-time aerosol mass spectrometer (AMS) data provides a method to obtain the total amount, overall fraction contribution, and properties of IEPOX-SOA (Robinson et al., 2011; Budisulistiorini et al., 2013; Chen et al., 2015). The $C_5H_6O^+$ ion at *m/z* 82 in AMS spectra, arising from decomposition and ionization of molecular IEPOX-SOA species, has also been suggested as a proxy for real-time estimation of IEPOX-SOA (Hu et al., 2015).



Heterogeneous reaction of OA with hydroxyl radicals (OH) is a contributor to aerosol
aging and significantly influences aerosol lifetime (George and Abbatt, 2010; George et al.,
2015). To describe the aging process, OA reaction rate coefficients with OH radicals ($k_{OH}$), or
alternatively uptake coefficients of OH ($\gamma_{OH}$), defined as the fraction of OH collisions with a
compound that result in reaction, have been reported for numerous laboratory studies. Values of
effective $\gamma_{OH}$ ($\leq 0.01$ to $\geq 1$) also can vary significantly under different reaction conditions, such
as different OA species (George and Abbatt, 2010), temperature and humidity (Park et al., 2008;
Liu et al., 2012; Slade and Knopf, 2014), OH concentrations (Slade and Knopf, 2013; Arangio et
al., 2015), and particle phase state or coatings (McNeill et al., 2008; Arangio et al., 2015). Most
of the studies that have reported $k_{OH}$ and $\gamma_{OH}$ are based on laboratory experiments, with few
experimental determinations of $k_{OH}$ based on field measurements under ambient conditions
(Slowik et al., 2012; Ortega et al., 2015), while no $\gamma_{OH}$ has been reported based on field studies to
our knowledge.
During the Southern Oxidant and Aerosol Study (SOAS), 17% of ambient OA was
estimated to be IEPOX-SOA (Hu et al., 2015). In this study, ambient gas and aerosol species
were sampled through an oxidation flow reactor (OFR) and a thermodenuder (TD) to investigate
heterogeneous oxidation and evaporation of ambient IEPOX-SOA, respectively. These systems
included an AMS and other on-line instruments measuring both gas and aerosol species inflow
and outflow. A simplified box model is used to investigate the fate of gas-phase IEPOX under
ambient and OFR conditions. The potential of evaporation to impact the lifetime of IEPOX-SOA
was evaluated. The heterogeneous reaction rate coeffcient ($k_{OH}$) and OH uptake coefficient ($\gamma_{OH}$)
of IEPOX-SOA with OH radicals are estimated from the OFR data. IEPOX-SOA aging during
the dry season of 2014 in central Amazonia as part of the Green Ocean Amazon
(GoAmazon2014/5, IOP2) experiment, using the same OFR experimental setup, was compared
to the SOAS results.
**2 Experimental method**
**2.1 Background and instrumentation**
The SOAS study (hereafter refer to "SE US study") took place in the SE US in the summer
(June 1– July 15) of 2013. Results shown here are from the SEARCH Centreville Supersite
(CTR) in a mixed forest in Alabama (32.95° N, 87.13°W; Hansen et al., 2003). The average





(±standard deviation) temperature and relative humidity (RH) of ambient air were 25±4℃ and
83±18%, respectively (Fig. S1). Biogenic volatile organic compounds (BVOCs) were highly
abundant with average isoprene and monoterpene concentrations of 3.3±2.4 ppb and 0.7±0.4
ppb, respectively, and they displayed clear diurnal variations (Fig. S1). Isoprene showed a broad
mid-afternoon peak (~5.8 ppb), and monoterpenes peaked during the nighttime and early
morning (~0.9-1.0 ppb). Chemically-resolved mass concentrations of submicron non-refractory
aerosol ($PM_1$) were measured by a high-resolution time-of-flight AMS (HR-ToF-AMS,
Aerodyne Research Inc., DeCarlo et al., 2006) at a time resolution of 2 min. Detailed information
about AMS setup, operation and data analysis is given in the supporting information and as well
as in Hu et al. (2015).

A "Potential Aerosol Mass" oxidation flow reactor (OFR) was used to investigate OA

formation/aging from ambient air over a wide range of OH exposures ($10^{10}$-$10^{13}$ molec. $cm^{-3}$ s).
This field-deployable OFR provides a fast and direct way to investigate oxidation processes of
ambient gas and aerosol with OH radicals under low-NO chemistry  (Kang et al., 2007; Lambe et
al., 2011; Li et al., 2015a; Ortega et al., 2015; Peng et al., 2015b; Palm et al., 2016). The OFR is
a cylindrical vessel (~13 L) with an average residence time of ~180-220 s in this study,
depending on the flow rates of sampled ambient air (3.5-4.2 L $min^{-1}$) (Fig. S2-S3). In the
"OFR185" method of OH production used in this study, two low-pressure mercury lamps inside
the OFR produce UV radiation at 185 and 254 nm (Peng et al., 2015b). OH radicals were
generated when the UV light initiated $O_2$, $H_2O$, and $O_3$ photochemistry (Li et al., 2015a). A large
range of OH exposures ($10^{10}$-$10^{13}$ molec. $cm^{-3}$ s) can be achieved by varying UV light intensity,
equivalent to several hours to several weeks of photochemical aging of ambient air (assuming a
24-hr average OH=$1.5×10^6$ molec. $cm^{-3}$; Mao et al., 2009). OH exposures in the OFR were
calculated by the real-time decay of CO added to the ambient air in the OFR (1-2 ppm; OH
reactivity≈5-10 $s^{-1}$). OH exposures in the OFR were calculated by the real-time decay of CO
added to the ambient air in the OFR (1-2 ppm; OH reactivity≈5-10 s-1). The empirical estimation
of OH exposure based on the OFR output parameters O3, water, and ambient OH reactivity (15
s-1) showed good agreement with that calculated from CO decay as shown in Fig. S4 (2015a).
The uncertainty of calculated OH exposures in the OFR was estimated as 35% based on
regression analysis (Li et al., 2015a; Peng et al., 2015b).



The average wall loss corrections for OA in OFR during the SE US study is 2±0.7%. This
wall loss is estimated by comparing the ambient OA concentrations to those concentrations after
the OFR when the UV lights were off and no oxidant was present (other than ambient $O_3$).
A TD was used to investigate the volatility of ambient OA and IEPOX-SOA. The
temperature in the TD increased linearly during the heating period (from 30℃ to 250℃ over 60
min) and then cooled down to 30℃ for 60 min. More detailed information on the TD technique
and instrumentation can be obtained elsewhere (Faulhaber et al., 2009; Huffman et al., 2009a;
Huffman et al., 2009b).
A typical sampling cycle during SE US study took a total of 24 min, sequentially sampling
ambient (4 min), TD (4 min), ambient (4 min), OFR with OH radicals as oxidant (4 min),
ambient (4 min), and OFR with other types of oxidation (e.g., $O_3$ or $NO_3$ as oxidants; 4 min), as
illustrated in the diagram in Fig. S2. Only OFR data for OH oxidation using OFR 185 method is
presented here. UV light intensities in the OFR were changed immediately after sampling the
second OFR outflow for each cycle. Thus, oxidant concentrations in the OFR had sufficient time
(at least 12 min, i.e. 3-4 flow e-folding times) to stabilize before the next OFR sampling interval.
The air from each sampling mode was sampled by the AMS, a scanning mobility particle sizer
(for measuring particle number size distributions; SMPS, TSI Inc.), and several other instruments
to measure related gas phase species, e.g., VOCs from proton-transfer-reaction mass
spectrometer (PTR-MS), $O_3$, CO and $H_2O$ (Table S1).
Measurements collected during the second Intensive Operating Period (IOP2) of the Green
Ocean Amazon (GoAmazon2014/5, hereinafter "Amazon study") Experiment (Martin et al.,
2015), which took place in the dry season of central Amazonia, are also presented here. The
region has high isoprene and monoterpene emissions (Karl et al., 2007; Martin et al., 2010). In
this analysis, data from the "T3" ground site (3.213 S, 60.599 W), a rural location 60 km west of
Manaus (Pop. 2 million) in the dry season (Aug. 15 to Oct. 15, 2014) are also shown. Unlike SE
US study, the aerosols in dry season of Amazon study were heavily influenced by biomass
burning, thus providing a difference dataset to investigate IEPOX-SOA heterogeneous reaction.
The instrument setup, OFR settings, sampling schemes and data processing were similar to those
for SE US study.
**2.2 IEPOX-SOA identification**



We classified ambient OA using positive matrix factorization (PMF) on the time series of
peak-fitted, high-resolution organic spectra measured by the AMS (Ulbrich et al., 2009). A factor
corresponding to ambient IEPOX-SOA was assigned based on its spectral features (e.g.
prominent $C_5H_6O^+$ ion at $m/z$ 82), and strong correlation with hourly or daily-measured 2-
methyltetrols (R=0.79), an oxidation product of isoprene oxidation via the IEPOX pathway
(Surratt et al., 2010; Hu et al., 2015), as well as with sulfate (R=0.75), which facilitates IEPOX-
SOA formation through direct reactions or nucleophilic effects (Nguyen et al., 2014a; Liao et al.,
2015). Unconstrained PMF analysis often fails when the factor fractions become too small
(<5%), e.g., as is for the IEPOX-SOA at higher OH exposures in the OFR in this study (Ulbrich
et al., 2009). To overcome this, a more advanced algorithm, the Multilinear Engine (ME-2)
(Paatero, 2007; Canonaco et al., 2013), was applied through the recently implemented Source
Finder (SoFi, Canonaco et al., 2013). In SoFi, the mass spectrum of the IEPOX-SOA factor was
constrained based on the ambient spectrum of IEPOX-SOA from conventional PMF, and the
concentrations of IEPOX-SOA factors were retrievable even at low concentrations. More
information can be found in Supp. Info. (Sect. 2 and Fig. S5-S9). Here after we will call IEPOX-
SOA PMF factor to be IEPOX-SOA for abbreviation.
In this study, $C_5H_6O^+$ data directly measured from AMS is used as a complementary tool to
examine/interpret the analysis results from IEPOX-SOA PMF factor, since both lab and ambient
results have shown $C_5H_6O^+$ is a very good tracer for IEPOX-SOA (Hu et al., 2015). Analyzing
$C_5H_6O^+$ is an easy alternative method to evaluate the physicochemical evolution of IEPOX-SOA,
that avoids the uncertainties related to PMF analysis, and thus provides further confidence in the
results. This is especially true when periods where the OA is dominated by IEPOX-SOA are
analyzed.
**2.3 Box model to simulate gas-phase IEPOX**
The chemistry of OH oxidation in the OFR is typical of low-NO conditions with $HO_2$ being
the dominant reaction partner of $RO_2$ radicals due to the greatly elevated $HO_2$ concentrations and
the very short lifetime of NO and $NO_x$ in OFR (Li et al., 2015a; Peng et al., 2015b). A box model
(KinSim 3.2 in Igor Pro. 6.37) was used to simulate the fate of gas-phase IEPOX under both
ambient and OFR conditions, as shown in Fig. 3 (Paulot et al., 2009; Xie et al., 2013; Bates et
al., 2014; Krechmer et al., 2015). A detailed description, including reactions and parameters in
the model, pH-dependent uptake coefficient of IEPOX onto aerosols ($\gamma_{IEPOX}$), aerosol surface



area calculations and estimated photolysis of IEPOX, can be found in Supp. Info. Section 3
(Table S2-3 and Fig. S10-14).

**3 Results and discussion**

**3.1 Low Volatility of IEPOX-SOA**

TDs are widely used to investigate the volatility distribution of OA in ambient air (e.g.
Faulhaber et al., 2009; Cappa and Jimenez, 2010). IEPOX-SOA evaporates more slowly upon
heating (Fig. 1a) than total OA over a very wide range of TD temperatures (<170℃), indicating
that IEPOX-SOA has a lower volatility than bulk OA. Consistent with that result, a lower
volatility of the IEPOX-SOA tracer $C_5H_6O^+$ in both SE US and Amazon studies was also found
(Fig. 2).
The volatility distributions of IEPOX-SOA and OA were estimated following the method of
Faulhaber et al. (2009), based on calibration of the relationship between TD temperature and
organic species saturation concentration at 298 K ($C^*$). Similar methods have been developed for
other thermal desorption instruments (e.g., Chattopadhyay and Ziemann, 2005; Lopez-Hilfiker et
al., 2016). The volatility distribution of IEPOX-SOA (Fig. 1b) shows mass peaks at $C^*=10^{-4} -$
$10^{-3}$ µg m$^{-3}$, which are much lower than those of diesel POA ($C^*=10^{-2}$ -1 µg m$^{-3}$) and biomass-
burning POA ($C^*=10^{-2}$-100 µg m$^{-3}$, Fig. 1d) at various OA concentrations (1-100 µg m$^{-3}$). Those
types of OA are reported to be semivolatile (Cappa and Jimenez, 2010; Ranjan et al., 2012; May
et al., 2013). The estimated distribution implies that very little of the ambient IEPOX-SOA was
actively partitioning to the gas phase during SE US study (Fig. 1b). Although we cannot rule out
some chemical changes during TD heating, this conclusion is dictated by the data at the lowest
TD temperatures, when such chemistry is less likely. Lopez-Hilfiker et al. (2016) have shown
that oligomer decomposition for IEPOX-SOA upon heating at ~90℃ was important during SE
US study, but that process will only make the measured volatility of IEPOX-SOA in TD higher
than it should be. This reinforces our conclusion about the low volatility of ambient IEPOX-
SOA, consistent with the independent results of Lopez-Hilfiker et al. (2016).
Several molecular species (e.g., 2-methyltetrols, $C_5$-alkene triols, IEPOX organosulfate and
its dimer) comprising IEPOX-SOA have been characterized both in field and chamber studies
(Surratt et al., 2010; Lin et al., 2012; Budisulistiorini et al., 2013; Liao et al., 2015). At the CTR
site during the SE US study, 2-methyltetrols, $C_5$-alkene triols and IEPOX organosulfate



measured by GC/MS and LC/MS in the particle phase accounted for an average of 80%
(individually 29%, 28% and 24%, respectively) of total IEPOX-SOA factor mass (Hu et al.,
2015). The volatilities of these IEPOX-SOA molecular species was estimated based on SIMPOL
group contribution method (Pankow and Asher, 2008). The species reported to comprise most of
IEPOX-SOA have relatively high $C^*$ (2-methyltetrol=2.7 µg m$^{-3}$; $C_5$-alkene triols=400 µg m$^{-3}$,
and IEPOX organosulfate=0.5 µg m$^{-3}$). The alkene triols in ambient air during SE US study
(where average OA mass concentration was 4.8 µg m$^{-3}$) should have been almost completely in
the gas phase (>98%), while 36% and 10% of the methyltetrol and organosulfate should have
been in the gas-phase, respectively. The $C^*$ of those monomer species is much higher than for
the bulk IEPOX-SOA ($C^* = 10^{-6}$ -$10^{-2}$ µg m$^{-3}$) that they are thought to comprise. On the other
hand, the estimated $C^*$ of a hypothetical methyltetrol molecular dimer (~$10^{-7}$ µg m$^{-3}$) is
significantly lower than that of most of the bulk IEPOX-SOA (Fig. 1d). This suggests that
IEPOX-SOA may exist as oligomers in the aerosol phase, but that the oligomers were not
evaporating as oligomers, rather decomposing and evaporating as monomer species at
temperatures intermediate with those corresponding to the $C^*$ of the monomers and the dimers,
consistent with results of Lopez-Hilfiker et al. (2016).

Further evidence supporting low volatility and strong oligomerization of IEPOX-SOA

molecular species has also been reported. Lin et al. (2014) showed oligomers as part of IEPOX-
SOA in filter-based LC/MS measurement at three sites (including CTR) during SE US study.
Some of the oligomers were separated by mass units of 100 ($C_5H_8O_2$) and 82 ($C_5H_6O$), which
would be consistent with $C_5$-alkene triol ($C_5H_{10}O_3$) and methyltetrol ($C_5H_{12}O_4$) oligomerization
though dehydration reactions (-$H_2O$ or 2 $H_2O$), or with other reactions resulting in similar
products. Results from online gas-particle partitioning measurements at the same site during this
study have shown that the measured particle-phase fractions ($F_p$, negatively correlated with $C^*$)
of ambient IEPOX-SOA tracers (e.g., 2-methyltetrols and $C_5$-alkene triols) are much higher than
expected based on the species vapor pressures, consistent with these tracers being formed during
GC analysis by decomposition of larger molecules (likely oligomers) (Isaacman-VanWertz et al.,
2016). Thus, the low volatility of IEPOX-SOA estimated from our TD data is consistent with
multiple other measurements.



Using the volatility distributions determined from the TD, the fractional losses for both OA

and IEPOX-SOA due to evaporation upon dilution can be estimated. This parameter can be
quantified as (Cappa and Jimenez, 2010):
$$E_{loss} = 100\% \left[ 1 - \frac{C_{OA}(DF)}{C_{OA}(0)/DF} \right]. \tag{1}$$

where $E_{loss}$ is the fractional OA loss due to evaporation; $C_{OA}(0)$ is the initial organic mass
concentration before dilution, and $DF$ is the dilution factor applied. $C_{OA}(DF)$ is the OA
concentration in equilibrium after dilution. Dilution factors varying from one to thirty were used
here. The results are shown in Fig. 1c. After a 30-fold dilution, IEPOX-SOA mass loss due to
evaporation is estimated to be ~5%, substantially lower than for total OA (17%). There are two
uncertainties affecting this result. One is that the real volatility distribution of IEPOX-SOA is
likely even lower, since the TD results are thought to be affected by oligomer decomposition
upon heating. The other one is that this calculation neglects the effect of possible decomposition
of oligomers onto monomers in ambient air. If that process occurs on a timescale of e.g., 1day, it
would lead to higher evaporated fractions than estimated here. The residence time of TD isis
~10-15s, which may not be sufficient time for oligomer decomposition, especially at the lower
temperatures that determined the upper end of the estimated volatility distribution. E.g. Vaden et
al (2011) reported that it took 24 h to evaporate 75% of α-pinene SOA. The kinetics of oligomer
decomposition of IEPOX-SOA under ambient conditions should be further investigated to fully
constrain its evaporation dynamics.

**283  3.2 Fate of gas-phase IEPOX**

IEPOX-SOA loadings exhibited a continuous decrease as OH exposure increases in the
OFR. To interpret the observed decay of IEPOX-SOA in the OFR, we first need to understand
whether additional IEPOX-SOA was formed in the OFR during SE US study. More details about
the IEPOX-SOA decay will be discussed in Sect. 3.3. Here, the box model described above (Fig.
3) was used to simulate the fate of gas-phase IEPOX in OFR and ambient conditions, as shown
in Fig. 4.
In ambient air, gas-phase IEPOX will either react with OH radicals to form more oxidized
gas-phase products (e.g. hydroxyacetone) (Bates et al., 2014; Bates et al., 2015), be taken up
onto acidic aerosol (Surratt et al., 2010), or be lost from the atmosphere by dry or wet deposition
(Nguyen et al., 2015). Photolysis of IEPOX in ambient air should be negligible, since the



epoxide and hydroxyl groups in IEPOX are photostable at visible or actinic UV wavelengths
(Fleming et al., 1959). A model scenario accounting for organic resistance with slower IEPOX
uptake than pure inorganic is applied to simulate the fate of gas-phase IEPOX. This scenario is
the most realistic assumption, since 67% of ambient aerosol is OA during SE US study (Fig.
S15). Results from an alternative model assuming pure inorganic aerosols are shown in Supp.
Info. The model predicts that the main pathway of gas-phase IEPOX removal in ambient air is
aerosol-phase uptake during SE US study, where about 75% of IEPOX was taken up by the
aerosol after one day under ambient conditions, because of the efficient uptake of gas-phase
IEPOX onto acidic ambient aerosols (pH=0.8±0.5) at the CTR site ($\gamma_{IEPOX}$=0.009, lifetime~1.8
h). The rest of IEPOX was lost to dry deposition to the surface (16%), according to reported
boundary layer of 1200 m and dry deposition rate of 3 cm s$^{-1}$ (Nguyen et al., 2015), or to gas-
phase reaction with OH (9%).
The fate of IEPOX sampled into the OFR differed from its fate in ambient air. Remaining
unreacted and then leaving OFR or destruction in the gas phase completely dominate the fate of
IEPOX under OFR conditions (Fig 4b). Negligible amounts of IEPOX (<1%) were taken up into
the aerosol phase in the OFR. This is mainly because the lifetime of IEPOX aerosol uptake
($\gamma_{IEPOX}$=0.002; lifetime=7.0h) was much longer than the OFR residence time (200s). The lower
$\gamma_{IEPOX}$ in OFR (0.002) than in ambient condition (0.008) was because of the higher pH of
aerosol leading to a slower IEPOX uptake. Higher pH in OFR (1.35±0.6) than that in ambient
(0.8±0.5) was because extra neutralized inorganic aerosol was formed in OFR. Photolysis of
IEPOX in OFR is estimated to be very minor (less than 0.2%) (Fig. 4b and Table S3). Loss of
IEPOX to the reactor walls is thought to be minor under the conditions of SE US study, given its
high vapor pressure (Krechmer et al., 2015; Palm et al., 2016).
IEPOX-SOA mass concentrations formed in both ambient and OFR conditions were
calculated as a function of OH exposure. For this estimate the molar mass of IEPOX-SOA and
the SOA molar yield ($\varphi_{SOA}$) of IEPOX, defined as the sum of formed aqueous phase SOA tracer
relative to the heterogeneous rate of gas-phase epoxide loss to particles (Riedel et al., 2015), are
needed. Using the measured molecular composition of IEPOX-SOA (Hu et al., 2015), and
assuming all species were present as dimers as discussed above, yields an average molar mass of
bulk IEPOX-SOA of 270 g mol$^{-1}$. Laboratory uptake experiments showed the SOA molar yield
of IEPOX is around 10-12% for acidic NH$_4$HSO$_4$ (Riedel et al., 2015). A molar mass of 270 g




325 mol$^{-1}$ and $\varphi_{SOA}$=6% (to account for the dimerization) for IEPOX-SOA were applied here. In the

326 OFR, the maximum modeled IEPOX-SOA mass concentrations were less than 12 ng m$^{-3}$,

327 peaking at ~1 day OH exposure. The model-predicted IEPOX-SOA formation is equivalent to

328 ~1% of the ambient IEPOX-SOA, indicating negligible IEPOX-SOA was formed in the OFR.

329 An upper limit of ~6% of the ambient IEPOX-SOA mass being formed in the OFR can be

330 derived assuming that the particles are 100% inorganic, as shown in Fig. S17.

331  In addition to the box model results, we also have experimental evidence demonstrating

332 negligible IEPOX-SOA formation in the OFR. During the Amazon study, standard additions of

333 isoprene (50-200 ppb) were injected into ambient air at the entrance of the OFR, during a period

334 when little SOA was formed from ambient precursors. After isoprene was exposed to varied OH

335 exposures (~10$^9$-10$^{12}$ molec. cm$^{-3}$ s) in the OFR in the presence of ambient aerosols, no

336 additional IEPOX-SOA formation was observed in the oxidized air exiting the OFR, as shown in

337 Fig. 5. Even under optimum OH exposures (8-11×10$^{10}$ molec. cm$^{-3}$ s), where most of the

338 isoprene and isoprene dydroxyhydroperoxide (ISOPOOH) are expected to be oxidized and

339 before substantial decay of IEPOX-SOA occurs, no enhancements of IEPOX-SOA tracer

340 $C_5H_6O^+$ ion abundance in OA spectra were observed. Consistent with our results, a laboratory

341 flow tube study (residence time = 1 min) of low-NO isoprene oxidation in the presence of

342 acidified inorganic seeds also reported negligible IEPOX-SOA formation (Wong et al., 2015).

343 Those results highlight a key limitation of this type of OFR: processes that do not scale with OH

344 and thus are not greatly accelerated in the reactor are not captured. This limitation can be

345 removed by seeding the OFR with H$_2$SO$_4$ particles, which greatly accelerate IEPOX aerosol

346 uptake. Simulation results (not shown) indicate that adding ~100 µg m$^{-3}$ of pure H$_2$SO$_4$ to the

347 ambient air allows to efficiently form IEPOX-SOA in the reactor.

348 **3.3 Lifetime of IEPOX-SOA against OH oxidation**

349  IEPOX-SOA loadings showed a continuous decrease as OH exposure increases in the OFR

350 (Fig. 6a). Since negligible IEPOX-SOA mass was added in the OFR (Sect. 3.2), this decay

351 should be due to the sum of all IEPOX-SOA loss processes. The loss of IEPOX-SOA is defined

352 empirically here as the loss of the molecular structures that result on AMS spectral features of

353 IEPOX-SOA (e.g., $C_5H_6O^+$ and $C_4H_5^+$ enhancements, Lin et al., 2012; Hu et al., 2015), such that

354 an IEPOX-SOA component cannot be distinguished in constrained PMF analysis. Evaporation,

355 photolysis and heterogeneous reaction with OH radicals are three possible loss pathways.



In principle some IEPOX-SOA could evaporate, if semivolatile molecules in equilibrium
with it were oxidized by OH. As discussed above, IEPOX-SOA itself has low volatility and only
a small fraction (~5%) may evaporate to the gas phase after dilution of a factor of 30. Even the
oligomer decomposition could be fast in the ambient air, this process will be negligible in the
flow reactor on a time scale of 3 min. Thus IEPOX-SOA evaporation is unlikely to contribute to
the large observed IEPOX-SOA loss (up to 90%).

Photolysis of IEPOX-SOA also cannot explain the large deceases of IEPOX-SOA in Fig.

6a. Washenfelder et al. (2015) reported that IEPOX-SOA during SOAS contributed negligibly to
the aerosol absorption at 365 nm. Lin et al. (2014) reported a wavelength-dependent effective
mass absorption coefficient (MAC)  value of ~247 $cm^2$ $g^{-1}$ at 254 nm for laboratory-generated
IEPOX-SOA on acidified $NH_4HSO_4$ seed. Using the MAC trend vs. wavelength and the
measured data down to 200 nm we estimate an MAC of ~5200 $cm^2$ $g^{-1}$ at 185 nm. Using those
absorption efficiencies (and assuming an upper limit quantum yield of 1) we can derive an upper
limit photolysis fraction of 1.5% of IEPOX-SOA in the OFR when neglecting other competing
effects (e.g. OH oxidation, Table S3 and Fig. S18). In addition, the actual quantum yield may be
much less than 1, as IEPOX-SOA molecular species contain mainly hydroxyl and carbonyl
groups (Surratt et al., 2010; Lin et al., 2014). Interactions between these groups are thought to
result in low quantum yields in the condensed phase (Phillips and Smith, 2014; Sharpless and
Blough, 2014; Peng et al., 2015a; Phillips and Smith, 2015). Therefore photolysis of IEPOX-
SOA should contribute negligibly to the observed IEPOX-SOA decay.

The observed decay of IEPOX-SOA in Fig. 6a must then be the result of heterogeneous

reactions with OH radicals. This process can be quantitatively described as:
$$IEPOX - SOA_i/IEPOX - SOA_0 = e^{-k_{OH} \times OH_i \times \Delta t_i} = e^{-k_{OH} \times OH_{exp,i}} \qquad (2)$$
where IEPOX-$SOA_i$ is the IEPOX-SOA mass concentration after the $i^{th}$ OH exposure step in the
OFR. IEPOX-$SOA_0$ is the initial ambient IEPOX-SOA entering the OFR; IEPOX-$SOA_i$/IEPOX-
$SOA_0$ is the mass fraction remaining of IEPOX-SOA in the OFR output, shown on Fig. 6a. $OH_i$
is the average OH concentration of step $i$ in the OFR, $\Delta t_i$ is the photochemical age.  $OH_{exp,i}$
$=OH_i \times \Delta t_i$ is the OH exposure of step $i$. $k_{OH}$ is the heterogeneous reaction rate coefficient
between IEPOX-SOA and OH radicals.

Fitting the results in Fig. 6a with Eq. (2) results in a $k_{OH}$ of $4.0\pm2.0\times10^{-13}$ $cm^3$ $molec.^{-1}$ $s^{-1}$

The $1\sigma$ uncertainty was obtained by Monte Carlo simulation, from propagation of the errors of





IEPOX-SOA$_i$/IEPOX-SOA$_0$ (9%) and the uncertainty of OH exposure (35%, Fig. S4). The
uncertainty of IEPOX-SOA$_i$/IEPOX-SOA$_0$ was estimated as 9% from PMF analysis of OFR data
(Hu et al., 2015).
A similar $k_{OH}$ value ($4.6 \times 10^{-13}$ cm$^3$ molec.$^{-1}$ s$^{-1}$) was obtained by fitting the IEPOX-SOA
tracer $C_5H_6O^+$ ion decay as a function of OH exposure during a period (June 26$^{th}$, 14:00-19:00)
when 80-90% of ambient OA was composed of IEPOX-SOA (Fig. S19-S20), which confirms the
$k_{OH}$ determined above.
For comparison, the average mass fraction remaining of IEPOX-SOA vs. OH exposure
during the Amazon study is also shown in Fig. 6a. A similar $k_{OH}$ value of $3.9 \pm 1.8 \times 10^{-13}$ cm$^3$
molec.$^{-1}$ s$^{-1}$ was obtained. Despite differences between the SE and Amazon studies, the similarity
of results from both studies increases our confidence in the derived value of the heterogeneous
reaction rate coefficient.
To investigate $k_{OH}$ of OA, multiple experiments (usually with RH<30%) with laboratory-
generated different types of OA have been conducted. The bulk of those OA in the lab usually
had mobility particle sizes ranging from 100-300 nm (Table 1), similar to that of IEPOX-SOA in
SE US (wet size=415 nm). The $k_{OH}$ value of IEPOX-SOA determined here is similar to
heterogeneous $k_{OH}$ determined in those laboratory studies, including highly-oxidized OA (e.g.
citric acid; $3.3 - 7.6 \times 10^{-13}$ cm$^3$ molec.$^{-1}$ s$^{-1}$) (Kessler et al., 2012), levoglucosan ($1.4 - 4.3 \times 10^{-13}$
cm$^3$ molec.$^{-1}$ s$^{-1}$) (Slade and Knopf, 2014), and pure erythritol ($2.5 \times 10^{-13}$ cm$^3$ molec.$^{-1}$ s$^{-1}$), which
has a similar structure to the 2-methyltetrols in IEPOX-SOA (Kessler et al., 2010). A summary
of $k_{OH}$ in this study and other laboratory studies with additional experimental information for
each study is shown in Table 1.
A dependence of $k_{OH}$ on ambient RH was found in both the SE US and Amazon studies,
with larger $k_{OH}$ at high RH, especially above 90% RH (Fig. 7). This effect may be due to higher
liquid water content, leading to a larger surface area that facilitates faster OH uptake to the
aerosol phase and thus resulting in faster $k_{OH}$ values (Slade and Knopf, 2014). Accounting for
liquid water content, the calculated particle surface areas show similar trends to $k_{OH}$, in both
studies, as shown in Fig. 7. The values of both parameters increase with RH (especially for
RH>90%).
An alternative explanation for the measured RH dependence would be the influence of
diffusion limitations. However, at the RH levels studied here (>40%), diffusion limitations of



OH in the aerosol phase are thought to be negligible (calculated lifetime of bulk diffusion of OH
radical < 1s). The diffusion coefficient of OH radical in liquid phase ($>10^{-14}\,\mathrm{m^2\,s^{-1}}$) was obtained
from other laboratory-generated OA (e.g. α-pinene derived SOA, levoglucosan particles)
(Renbaum-Wolff et al., 2013; Arangio et al., 2015; Li et al., 2015b). Li et al. (2015b) reported
that the diffusion of $NH_3$ on laboratory biogenic SOA is only slowed at much lower transition
RH (10-40%) than that for liquid/solid phase transition (50-80%). This supports that under the
conditions in SE and Amazon studies diffusion limitations should not play a role.
The ambient lifetime of IEPOX-SOA due to the heterogeneous reaction with OH radicals
was estimated to be more than 2 weeks (19±9 days) based on the average $k_{OH}$ ($4.0\pm2.0\times10^{-13}\,\mathrm{cm^3}$
$\mathrm{molec.^{-1}\,s^{-1}}$), assuming an average ambient OH concentration of $1.5\times10^6\,\mathrm{molec.\,cm^{-3}}$. A similar
lifetime can be estimated for the Amazon study. Longer lifetimes of 48 days in SE US study and
99 days in Amazon study were estimated when the observed average 24h OH concentration in
both studies ($0.6\times10^6\,\mathrm{molecule\,cm^{-3}}$ in SE US and $0.3\times10^6\,\mathrm{molecule\,cm^{-3}}$ in Amazon) were used
(Krechmer et al., 2015). The long lifetime of IEPOX-SOA against heterogeneous oxidation is
consistent with the estimated lifetime of total OA in urban and forested areas (Ortega et al.,
2015; Palm et al., 2016), and also pure highly-oxidized OA (1-2 weeks) in laboratory studies
(Kessler et al., 2010; Kessler et al., 2012).
**3.4 Fate of Oxidized IEPOX-SOA mass**
It is of interest to determine whether the mass of IEPOX-SOA continues to be present in
the aerosol after OH heterogeneous oxidation, albeit as a different chemical form, or whether it
evaporates from the particles. Functionalization reactions would favor the former, while
fragmentation reactions would favor the latter (George et al., 2007).
At lower OH exposures ($<1\times10^{12}\,\mathrm{molec.\,cm^{-3}\,s}$) during daytime, SOA formation (non-
IEPOX-SOA) was observed in the OFR (e.g., from monoterpene and sesquiterpenes oxidation,
Fig. 6b), making it difficult to discern whether functionalization or fragmentation dominated for
IEPOX-SOA losses. However, at OH exposures in the OFR above $1\times10^{12}\,\mathrm{molec.\,cm^{-3}\,s}$, net
SOA formation from ambient air was no longer observed. This is presumably due to organic
vapors undergoing multiple generations of oxidation and fragmenting in the gas phase in the
OFR (Palm et al., 2016). For that OH exposure range, changes of the aerosol phase should be
dominated by heterogeneous reactions. In this regime, OA mass was lost at a rapid rate of ~6 %
OA mass per $1\times10^{12}\,\mathrm{molec.\,cm^{-3}\,s}$ of OH exposure through volatilization. A very similar rate





was observed for the IEPOX-SOA (~7% per $1 \times 10^{12}$ molec. cm$^{-3}$ s), which implies that the main
loss mechanism of IEPOX-SOA at higher OH exposures is due to volatilization following
fragmentation. In the period when 80-90% of OA was composed of IEPOX-SOA, the OA also
showed an up to 70% mass loss (Fig. S20), confirming the conclusion that a high fraction of
IEPOX-SOA was volatilized to the gas phase after heterogeneous reaction at higher OH
exposures.

The aerosol mass losses of IEPOX-SOA and OA into gas phase are consistent with

laboratory experiments of heterogeneous reaction of pure erythritol particles (a surrogate of the
IEPOX-SOA tracer 2-methyltetrols, see Fig. 6b), which also showed that OH oxidation led to
formation of volatile products escaping to the gas phase (Kessler et al., 2010; Kroll et al., 2015).
We note however that IEPOX-SOA is mostly composed of oligomers, rather than monomers as
with erythritol.
**3.5 Estimation of reactive uptake coefficient (γ) of OH**

By quantifying the removal of IEPOX-SOA in the aerosol phase, an effective reactive

uptake coefficient of OH ($\gamma_{OH}$) on the aerosol in the OFR can be estimated. To our knowledge,
this is the first time that $\gamma_{OH}$ has been derived from measurements of ambient SOA aging.

The variable $\gamma_{OH}$ can be calculated from $k_{OH}$ per Smith et al. (2009):


$$\gamma_{OH} = \frac{4 \cdot k_{OH} \cdot V_{IEPOX-SOA} \cdot \rho_0 \cdot N_A}{\bar{c} \cdot S_{IEPOX-SOA} \cdot MW_{IEPOX-SOA}} = \frac{4 \cdot k_{OH} \cdot D_{surf} \cdot \rho_0 \cdot N_A}{\bar{c} \cdot MW_{IEPOX-SOA}},$$    (3)

where $k_{OH}$ is the heterogeneous reaction rate coefficient of IEPOX-SOA discussed above
($4.2 \pm 2.1 \times 10^{-13}$ cm$^3$ molec.$^{-1}$ s$^{-1}$); $\rho_0$ is density of aerosol in OFR, which is estimated as $1.46 \pm 0.49$
g cm$^{-3}$ based on the aerosol composition (Fig. S15). $N_A$ is Avogadro's number; $\bar{c}$ is the mean
speed of gas-phase OH radicals, calculated as $(8RT/\pi M)^{0.5}$ (R is the universal gas constant, $T$ is
the temperature in K, and $M$ is the molar mass of the OH radical). The calculated $\bar{c}$ for OH (at
293 K) is 604 m s$^{-1}$. $MW_{IEPOX-SOA}$ is the molar mass of IEPOX-SOA. The estimated
$MW_{IEPOX-SOA}$=270 g mol$^{-1}$ was used here, which is similar to isoprene-SOA molar mass of 252
g mol$^{-1}$ estimated from a separate flow tube study based on CCN measurement (King et al.,
2010). An uncertainty of 30% is assigned to $MW_{IEPOX-SOA}$. $V_{IEPOX-SOA}$ and $S_{IEPOX-SOA}$ are the
mean volume and surface areas of IEPOX-SOA. If we assume IEPOX-SOA is uniformly mixed
with the other aerosol species, and independent of particle size, then IEPOX-SOA will account





for x% of total aerosol volume ($V_{total}$) and x% of total aerosol surface area ($S_{total}$). The ratio
between $V_{IEPOX-SOA}$ and $S_{IEPOX-SOA}$ can be expressed as: x%×$V_{total}$/x%×$S_{total}$=$V_{total}$/$S_{total}$.
Values of x varied from 0-90 based on source apportionment results. However, the exact value of
x is not needed here since it eventually was canceled out in the calculation. For a spherical
particle $V_{total}/S_{total}$ equals to $d_{surf}$/6. $d_{surf}$ is defined as surface-weighted particle diameter.
The dried surface-weighted aerodynamic size distribution of *m/z* 82 (background corrected),
tracer of IEPOX-SOA (Hu et al., 2015), peaks around 400 nm (Fig. 8), which is equivalent to
mobility size of ~274 nm. By applying the average particle size growth factor of 1.5 calculated
from average kappa (0.27) and ambient RH (Nguyen et al., 2014b), the average $d_{surf}$ of wet
IEPOX-SOA is estimated as 410 nm; Similar method was applied to calculate $d_{surf}$ of wet
IEPOX-SOA in Amazon study, which is finally calculated to be 490 nm.

The average mass-weighted aerodynamic size distribution of *m/z* 82 in SOAS and Amazon

($d_{va}$=~500 nm and 600 nm) is consistent with that of sulfate ($d_{va}$=~450 nm and 510 nm), which
may indicate sulfate control of the IEPOX uptake formation pathway (Xu et al., 2014; Liao et al.,
2015; Marais et al., 2016). Both peaks of *m/z* 82 and sulfate were systematically larger than of
total OA ($d_{va}$=~370 or 400 nm), suggesting the IEPOX-SOA formation in SE US and Amazon
studies may be partially contributed by aqueous/cloud processing (Meng and Seinfeld, 1994).
The systematically higher oxidation level of IEPOX-SOA in the ambient air than from chamber
studies also support this conclusion (Chen et al., 2015; Hu et al., 2015).

Finally, $\gamma_{OH}$ is estimated as 0.59±0.33 under a range of OH concentrations between $10^7$-$10^{10}$

molec. cm$^{-3}$, which is consistent with the range of $\gamma_{OH}$ (0.37-0.77) calculated for highly oxidized
OA in laboratory studies (Table 1). The uncertainty of $\gamma_{OH}$ was estimated by MonteCarlo
simulation, propagated from errors of each parameter in equation (2) (50% for $k_{OH}$, 30% for $d_{surf}$,
28% for $\rho_0$, and 30% for $MW_{IEPOX-SOA}$). When considering the apparent RH effect on $k_{OH}$,
estimated $\gamma_{OH}$ varies between 0.34-1.19. The $\gamma_{OH}$ above 1 at the highest RH range (90-100%)
might be due to secondary reactions of IEPOX-SOA in the more dilute liquid phase. The
estimated $\gamma_{OH}$ in Amazon study is around 0.68±0.38.

Ambient particles in both SOAS and GoAmazon were liquid as quantified by particle

bounce experiments (Bateman et al., 2015; Pajunoja et al., 2016) and thus kinetic limitations to
OH uptake in the OFR should not play a role (Li et al., 2015). In this study, we calculated $\gamma_{OH}$
based on a wide range of OH concentrations ($10^7 - 10^{10}$ molec. cm$^{-3}$). Several laboratory





experiments suggest that OH uptake should obey the Langmuir–Hinshelwood (LH) kinetic
mechanism, where $\gamma_{OH}$ tends to lower under higher OH concentrations, because of a saturation of
surface reactive sites at higher OH concentrations (Che et al., 2009; George and Abbatt, 2010;
Slade and Knopf, 2013). We have calculated $k_{OH}$ at different OH exposure ranges ($10^{10}$ to $10^{11}$-
$10^{13}$ molec. cm$^{-3}$ s$^{-1}$, Fig. S22). No obvious OH dependence of $k_{OH}$ ($\gamma_{OH}$) was found above $3\times10^{9}$
molec. cm$^{-3}$ (beyond where $k_{OH}$ calculation is more robust), which suggests the $\gamma_{OH}$ calculated in
this study does not depend on OH concentration. It may be because in this study the OH uptake
happened in the liquid particle, causing OH diffusion not to be limited by surface dynamics.
More consideration of other factors (e.g., surface regeneration due to volatilization; aerosol
phase influence) should be explored in future studies of the $\gamma_{OH}$ for IEPOX-SOA.

## 4. Conclusions

We investigated volatility and aging process of IEPOX-SOA during the late spring and
early summer of SE US and the dry season of central Amazonia with field-deployed
thermodenuder and oxidation flow reactor. IEPOX-SOA had a volatility distribution much lower
than those of the monomer tracers that have been reported as comprising most of its mass. Much
of IEPOX-SOA likely exists as oligomers in the aerosol phase. The kinetics of decomposition of
oligomers to monomers needs further investigation to fully constrain the lifetime of IEPOX-SOA
against evaporation.
The formation of IEPOX-SOA in the field and in the OFR flow reactor was investigated. In
contrast to the efficient IEPOX uptake in the ambient air, negligible IEPOX-SOA was formed in
the OFR under OH oxidation, as the OFR as used here cannot accelerate processes such as
aerosol uptake and reactions that do not scale with OH. Simulation results indicate that adding
~100 µg m$^{-3}$ of pure $H_2SO_4$ to the ambient air would allow to efficiently form IEPOX-SOA in the
reactor. Photolysis and evaporation of IEPOX-SOA in the OFR contributed negligibly to
IEPOX-SOA loss. From the OFR results, we determined the lifetime of IEPOX-SOA through
heterogeneous reaction with OH radicals ($k_{OH}$=4.0±2.0×$10^{-13}$ cm$^{3}$ molecule$^{-1}$ s$^{-1}$ in SE US and
3.9±1.8×$10^{-13}$ cm$^{3}$ molecule$^{-1}$ s$^{-1}$ in the Amazon) is equivalent to more than a 2-week
photochemical aging lifetime (assuming OH = 1.5×$10^{6}$ molec. cm$^{-3}$). The mass lost at high OH
exposures is mainly volatilized, rather than transformed into other aerosol species with different
composition, which suggests fragmentation plays an important role during ambient aging
process.



Values of effective $\gamma_{OH}$ based on the measured IEPOX-SOA $k_{OH}$ and other particle
parameters were determined to be 0.59±0.33 in SE US and 0.68±0.38 in Amazon with no
dependence on OH concentration over the range $10^7$-$10^{10}$ molecule cm$^{-3}$. This is the first time of
$\gamma_{OH}$ was estimated based on ambient SOA. Positive correlation between $\gamma_{OH}$ and wet particle
surface areas (RH dependent) suggest that OH uptake is surface area-limited. The substantially
larger size distribution of IEPOX-SOA tracer $m/z$ 82 and sulfate vs. bulk OA suggests that
IEPOX-SOA formation in SE US study may be controlled by sulfate and/or influenced by cloud
processing. However, the effect of aqueous processing under very dilute conditions relevant to
clouds has not been investigated to our knowledge. Our results provide constraints on the sinks
of IEPOX-SOA, which are useful to better quantify OA impacts on air quality and climate.

**Acknowledgements**
This study was partially supported by EPRI-10004734, NSF AGS-1243354 and AGS-1360834,
NASA NNX15AT96G, DOE (BER/ASR) DE-SC0011105, and NOAA NA13OAR4310063.
BBP and JEK were partially supported by EPA STAR Fellowships (FP-91761701-0 & FP-
91770901-0). We thank Annmarie Carlton, Eric Edgerton, and Karsten Baumann for their
organization of the SOAS Supersite; Cassandra Gaston and Joel Thornton from the University of
Washington for advice in the use of their IEPOX uptake model; Jian Wang from Brookhaven
National Laboratory for advice on aerosol hygroscopicity during GoAmazon2014/5; Ying-Hsuan
Lin and Jason D. Surratt from the University of North Carolina for sharing their published MAC
data of IEPOX-SOA; Hongyu Guo and Rodney J. Weber from Georgia Institute of Technology
for providing their estimated pH for comparison to our pH calculation results; and John Crounse
and Paul Wennberg from Caltech for gas-phase IEPOX and ISOPOOH data in SOAS. This paper
has not been reviewed by EPA and no endorsement should be inferred. (A portion of) The
research was performed using EMSL, a DOE Office of Science User Facility sponsored by the
Office of Biological and Environmental Research and located at Pacific Northwest National
Laboratory. SEARCH network operations are supported by Southern Company and EPRI.







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



**Table 1** Summary of $k_{OH}$, $\gamma_{OH}$ and different experiment parameters used in this study and other
lab studies.

| Species Name | $k_{OH}\times10^{12}$ (cm$^3$ molec.$^{-1}$ s$^{-1}$) | $\gamma_{OH}$ | OH conc.(molec.cm$^{-3}$) | React. time | Particle Size (nm) | RH | References |
|---|---|---|---|---|---|---|---|
| IEPOX-SOA in SE US | 0.40±0.20 | 0.59±0.33 | $10^7-10^{10}$ | ~200s | 415 | ~83% | (1) |
| IEPOX-SOA In SE US RH dependent | 0.32 | 0.34 | $10^7-10^{10}$ | ~200s | 302 | <60% | (1) |
|  | 0.33 | 0.39 | $10^7-10^{10}$ | ~200s | 328 | 60-80% | (1) |
|  | 0.34 | 0.46 | $10^7-10^{10}$ | ~200s | 380 | 80-90% | (1) |
|  | 0.64 | 1.19 | $10^7-10^{10}$ | ~200s | 525 | 90-100% | (1) |
| IEPOX-SOA in Amazon | 0.39±0.19 | 0.68±0.38 | $10^7-10^{10}$ | ~200s | 490 | ~86% | (1) |
| IEPOX-SOA in Amazon RH dependent | 0.35 | 0.45 | $10^7-10^{10}$ | ~200s | 363 | <60% | (1) |
|  | 0.35 | 0.46 | $10^7-10^{10}$ | ~200s | 380 | 60-80% | (1) |
|  | 0.37 | 0.54 | $10^7-10^{10}$ | ~200s | 415 | 80-90% | (1) |
|  | 0.53 | 1.09 | $10^7-10^{10}$ | ~200s | 576 | 90%-100% | (1) |
| **Highly oxidized organic species** | | | | | | | |
| *BTA$^a$* | 0.76 | 0.51 | $\sim10^9-3\times10^{11}$ | ~37s | ~130-145 | 30% | (2) |
| *Citric acid* | 0.43 | 0.37 | $\sim10^9-3\times10^{11}$ | ~37s | ~130-145 | 30% | (2) |
| *Tartaric acid* | 0.33 | 0.40 | $\sim10^9-3\times10^{11}$ | ~37s | ~130-145 | 30% | (2) |
| *Erythritol* | *0.25* | *0.77* | $\sim1\times10^9-2\times10^{11}$ | ~37s | ~200 | 30% | (3) |
| **Motor oil particles** | | | | | | | |
| *Diesel particles* | 0.4-34 | 0.1-8 | $0.6-40\times10^6$ | 4h | ~300 | 10-75% | (4) |
| *Nucleated motor oil particles* | N/A | 0.72 | $0-3\times10^{10}$ | 37s | ~170 | ~30% | (5) |
| **Biomass burning tracers** | | | | | | | |
| *Levoglucosan* | 0.31 | 0.91 | $\sim1\times10^9-2\times10^{11}$ | ~37s | ~200 | 30% | (3) |
|  | 0.14-0.43 | 0.21-0.65 | $10^8$ to $10^9$ | N/A | 120-267 | 0-40% | (6) |
|  | N/A | 0.15-0.6 | $10^7-10^{11}$ | <1 s | N/A | 0% | (7) |
| *Abietic acid* | N/A | 0.15-0.6 | $10^7-10^{11}$ | N/A | N/A | 0% | (7) |
| *Nitroguaiacol* | N/A | 0.2-0.5 | $10^7-10^{11}$ | N/A | N/A | 0% | (7) |
| *MNC$^b$* | 0.04-0.16 | 0.07-0.22 | $10^8$ to $10^9$ | N/A | 203-307 | 0-26% | (6) |
| **Other pure organic species** | | | | | | | |
| *Squalene* | N/A | 0.3±0.07 | $1\times10^{10}$ | ~37s | ~160 | 30% | (8) |
| *Squalene* | 1.8-1.9 | 0.49-0.54 | $1-7\times10^8$ | 1.5-3h | ~220 | 30% | (9) |
| *Palmitic Acid* | N/A | 0.8-1 | $1.4-3\times10^{10}$ | 10-17s | 85-220 | ~16% | (10) |

$^a$ 1, 2, 3, 4-Butanetetracarboxylic acid; $^b$ 4-methyl-5-Nitrocatechol
(1) This study; (2) (Kessler et al., 2012); (3) (Kessler et al., 2010); (4) (Weitkamp et al., 2008); (5) (Isaacman et al., 2012); (6)
(Slade and Knopf, 2014); (7) (Slade and Knopf, 2013); (8) (Smith et al., 2009); (9) (Che et al., 2009); (10) (McNeill et al., 2008).





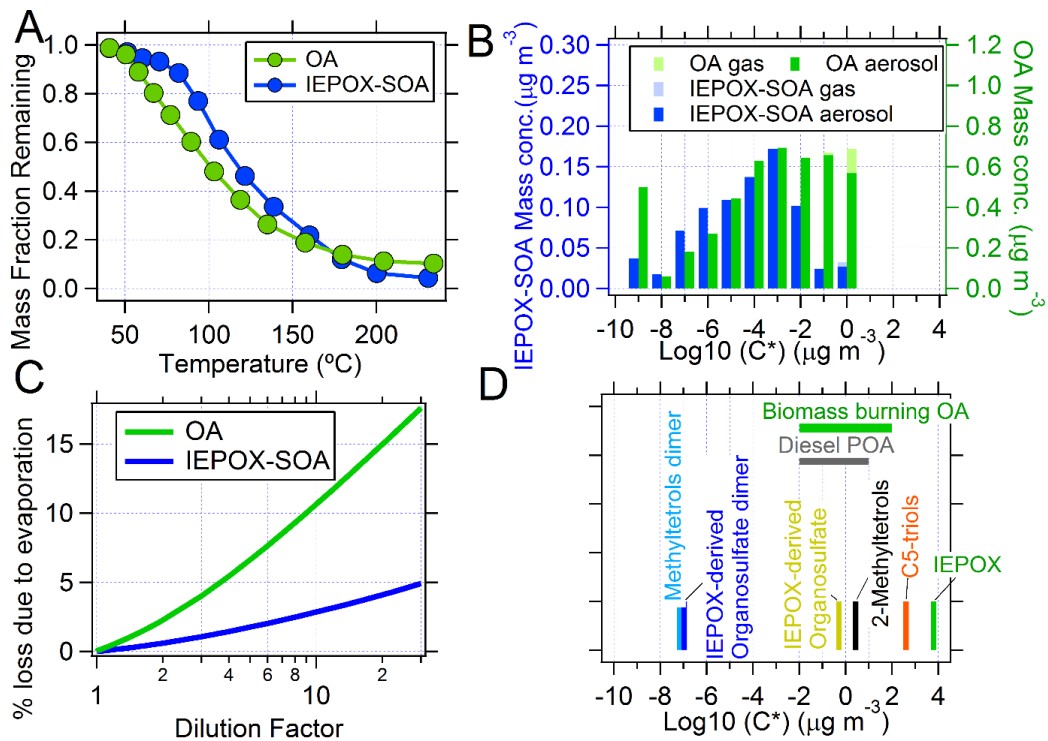


**Figure 1** (a) Mean mass fraction remaining of IEPOX-SOA and OA versus temperature in TD
("thermograms") during SE US study. (b) Volatility distributions of IEPOX-SOA and OA
estimated from TD thermograms (see text). Bars are offset for clarity and were both calculated
for integer log($C^*$) values. (c) Evaporation losses of IEPOX-SOA and OA as a function of
dilution factors. (d) Volatility of typical IEPOX-SOA molecular species in the aerosol phase
based the on SIMPOL group contribution method (Pankow and Asher, 2008). The reduction in
vapor pressure upon addition of a nitrate group was used to estimate the effect of the sulfate
group, due to lack of SIMPOL parameters for the latter, and the derived $C^*$ may be
overestimated for this reason.





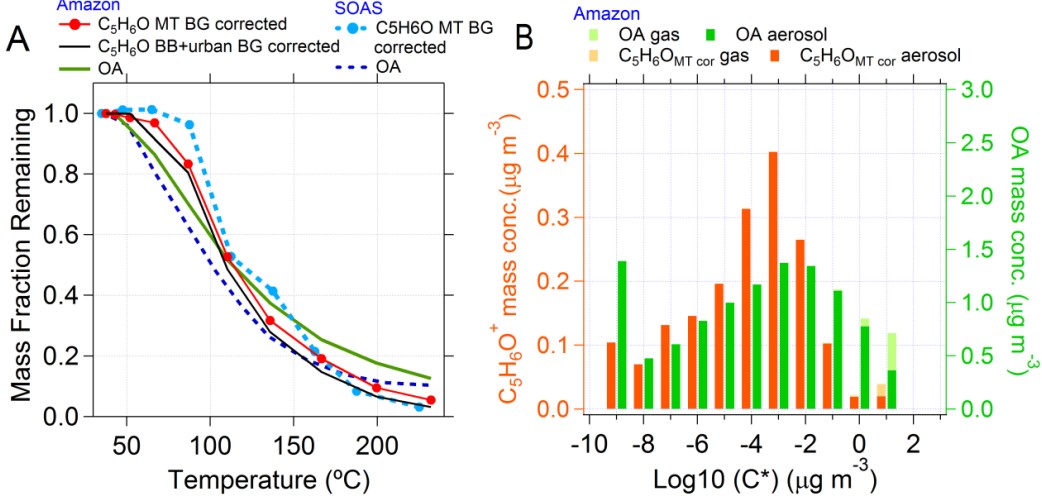


**Figure 2** (a) Thermogram of OA and background-corrected $C_5H_6O^+$ ion in the SE US and
Amazon studies. (b) Volatility distributions of $C_5H_6O^+$ and OA estimated based on TD
thermograms from the Amazon study.






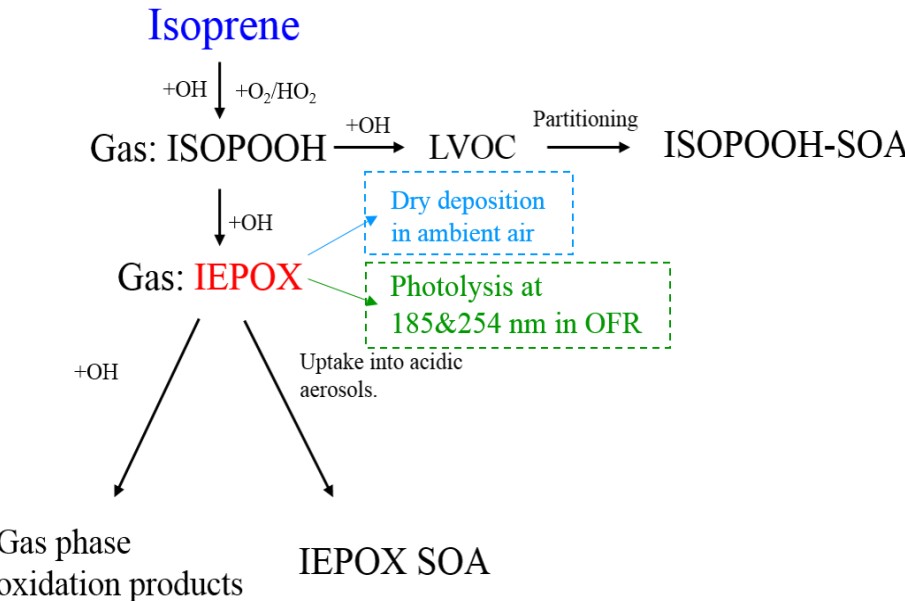


**Figure 3** Mechanism diagram of gas-phase IEPOX model in ambient and OFR conditions.
ISOPOOH-SOA is referred to SOA formed through gas-particle partitioning of low-volatile
VOCs from oxidation of isoprene 4-hydroxy-3-hydroperoxide (4,3-ISOPOOH) under low-NO
conditions (Krechmer et al., 2015).






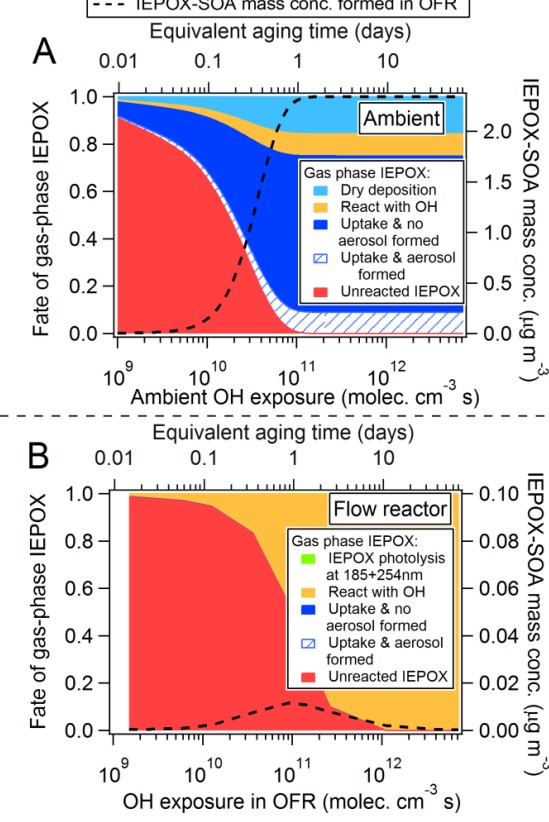


**Figure 4** Modeled IEPOX fate (a) in ambient air and (b) oxidation flow reactor (OFR)
conditions in SE US study. The uptake rate of gas-phase IEPOX onto aerosol is calculated by
using the model of Gaston et al. (2014), and is mainly influenced by aerosol pH (estimated as 0.8
and 1.35 for ambient and OFR aerosol, respectively) and aerosol surface areas (300 and 350
$\mu m^2/cm^3$ for ambient and OFR aerosol, respectively). The calculated IEPOX-SOA mass
concentrations are shown in Fig. 3. The OH exposures for both panels range 15 min-2 months of
atmospheric equivalent age (at OH concentration=$1.5\times10^6$ molec. cm$^{-3}$).

942



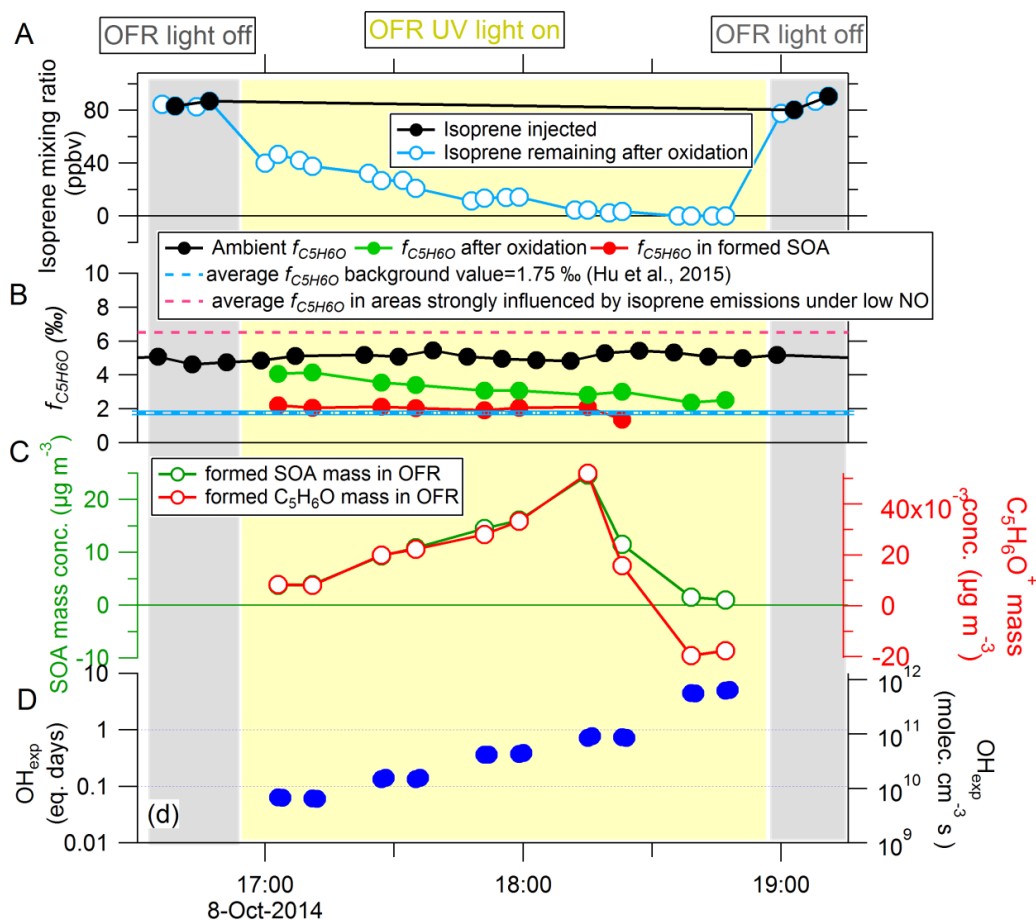

943

**Figure 5.** Isoprene standard addition experiment in ambient air during the GoAmazon2014/5
study. (a) Isoprene concentration injected and remaining after OFR. (b) Time series of ambient
$f_{C_5H_6O}$, $f_{C_5H_6O}$ in OA after oxidation and $f_{C_5H_6O}$ in newly formed SOA from OFR oxidation. The
average background value $f_{C_5H_6O}$=1.75‰ from urban and biomass burning emissions and
$f_{C_5H_6O}$=6.5 ‰ from aerosol strongly influenced by isoprene emissions are also shown (Hu et al.,
2015). (c) Time series of mass concentration of newly formed SOA (left axis) and $C_5H_6O^+$ (right
axis). (d) Time series of equivalent aging time (left axis) and OH exposure in OFR (right axis).
OH concentration=$1.5\times10^6$ molec. cm$^{-3}$ was assumed here to calculate equivalent OH aging
times. The grey background indicates OFR light off period and light yellow is OFR light on
period. Different OH exposures were achieved by varying the UV light intensity. Residence time
in the OFR was about 200 s.




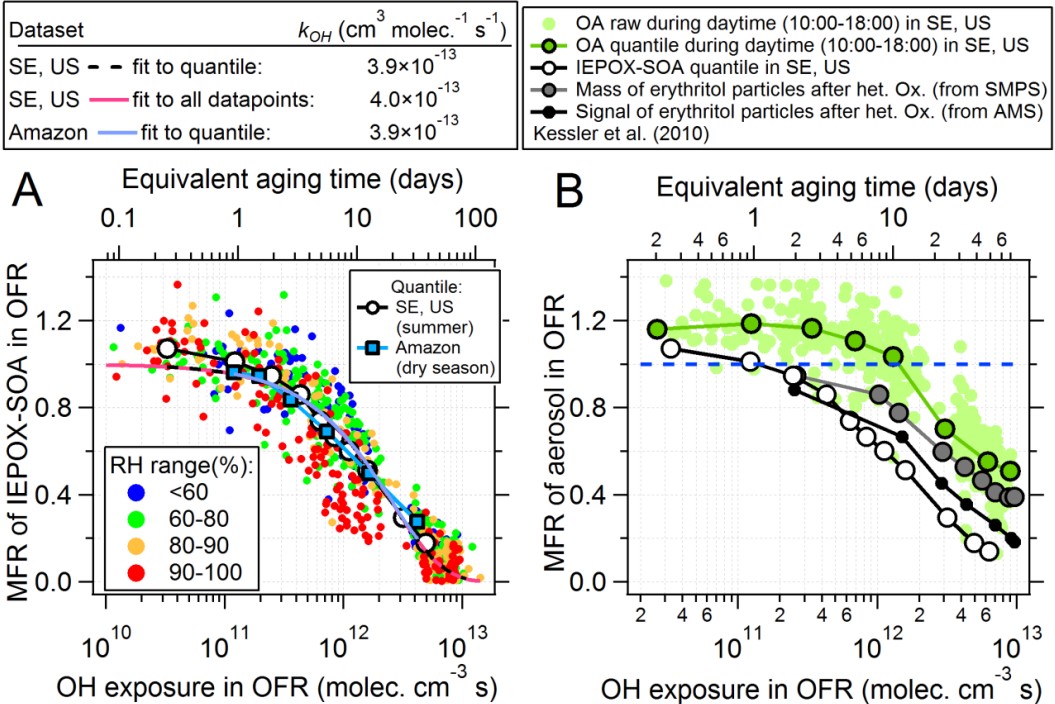


**Figure 6** (a) Mass fraction remaining (MFR) of IEPOX-SOA in OFR output as a function of OH
exposure during the entire SOAS and GoAmazon 2014/5 (dry season) studies. Individual
datapoints from SOAS are color-coded by ambient RH. Similar data for GoAmazon 2014/5 are
shown in Fig. S24. (b) Mass fraction of OA remaining in OFR output as a function of OH
exposure in daytime (12:00-18:00) during SOAS. Also shown is the MFR of pure erythritol
particles after heterogenous oxidation as detected by SMPS and by AMS for reference (Kessler
et al., 2010). Erythritol has a similar structure to the IEPOX-SOA tracers 2-methytetrols.




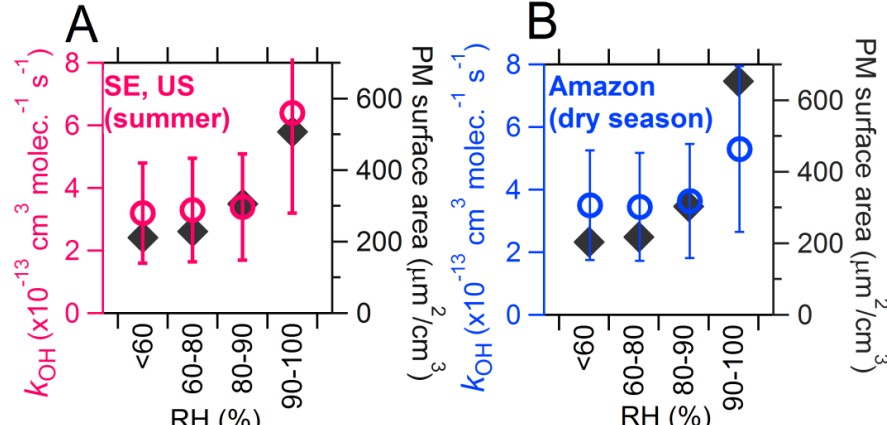



**Figure 7.** Estimated $k_{OH}$ of IEPOX-SOA vs. ambient RH during the SOAS and Amazon studies. The ambient wet particle surface areas in both studies are shown on the right axis.








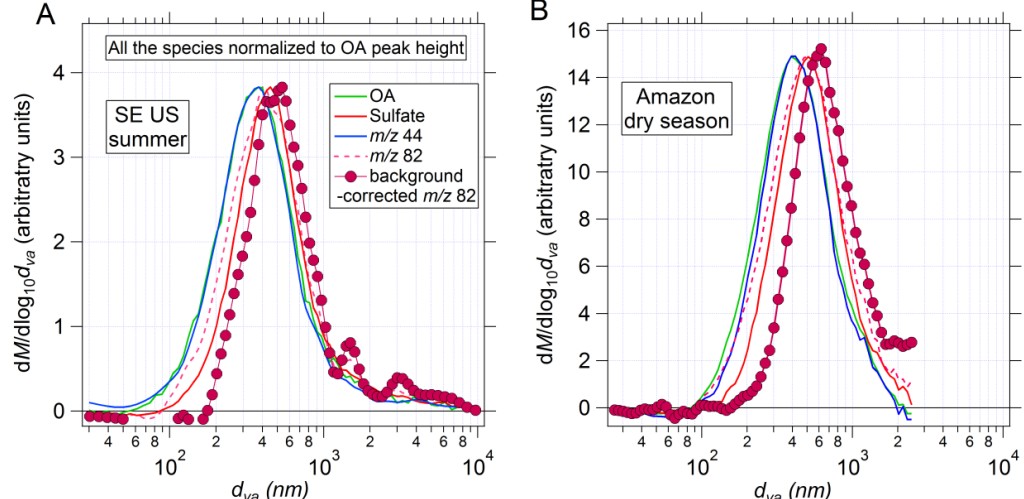


**Figure 8.** Average mass-weighted aerodynamic size distribution of OA, sulfate, *m/z* 44 and *m/z*
82 in (a) SE US and (b) Amazon. The mass size distribution of *m/z* 82 with background
correction is also shown. The background correction method was introduced in Hu et al.(2015).
Heights of all the size distributions are set to the same value for ease of visual comparison.