# Peer review of "Volatility and lifetime against OH heterogeneous reaction of ambient Isoprene Epoxydiols- 2 Derived Secondary Organic Aerosol (IEPOX-SOA)"

_Atmospheric Chemistry and Physics, 2016_

## Referee Comment (RC1) · Anonymous Referee #1 · 27 Jun 2016

This is a very interesting study about the volatility and lifetime of IEPOX-derived SOA. As the authors point out in the manuscript, studies on heterogeneous reactions of ambient SOA with hydroxyl radicals are missing. This work presents a well-conduct series of ambient measurements and box models that provide supporting evidences for their findings. The study is technically well done and the paper is well written. I think it is suitable for publication in ACP after few minor revisions. My major comment is on the determination of the IEPOX-derived SOA lifetime as discussed below.

Line 70: The authors should consider adding the references and not only cite their previous work. Add these references: Robinson et al., 2011; Budisulistiorini et al., 2013; 2015; Chen et al., 2015, Xu et al., 2015

Line 72: Add Riedel et al., 2015

Line 82: Similar comment as Line 70, add references: Robinson et al., 2011; Lin et al., 2012; Budisulistiorini et al., 2013

Line 85: Add reference: Kroll et al., 2015

Line 114: How did temperature affect the reactivity of IEPOX-derived SOA? Indeed as shown by Lai et al. (2015), small changes in the temperature (20C-30C) could impact OH heterogeneous oxidation.

Line 125: Why did the authors limit the range of OH exposures from 10e+10 – 10e+13 molecule cm-3? Indeed, as previously reported and discussed below, OH uptake and heterogeneous oxidation is impacted by the concentration of OH radicals (Arangio et al., 2015).

Line 154: The authors mentioned other type of oxidants, such as O3 and NO3. If they performed some experiments using different oxidants, it would be interesting to discuss the reactivity of these oxidants (especially NO3) and compare the results with OH radicals. Could the authors provide additional experimental information for the production of NO3 radicals in the OFR?

Line 155: The authors mentioned that they used different methods (i.e. OFR 185 and OFR 254?) but decided to present only results from one. Could they explain the reasons and add some information and/or comparison between both methods?

Line 168-170: The authors mentioned that during the dry season SOA were significantly influenced by biomass burning. This statement is supported by neither references nor experimental data. Additional information is needed to evaluate the importance of BBOA in OA. In addition, how do the authors expect the presence of BBOA to impact the IEPOX-SOA aging?

Line 192: Similar comment as line 82, add the references.

Lines 203-204: Riedel et al. (2016) have recently reported bulk laboratory measurements and determined the reaction rate constants of IEPOX-SOA products from the reactive uptake of IEPOX. The authors didn't consider this study and only use a global rate constant. The rates constants proposed by Riedel et al. (2016) should be added in the model and results from both approaches should be compared in the paper.

Lines 270-271: At which temperature did the authors perform these evaporation experiments?

Lines 295-296: Riva et al. (2016) have shown that IEPOX reactive uptake could be significantly reduced by the presence of organic on seed aerosols. The authors should add this reference and also discuss this potential impact in their model.

Lines 321-323: The authors should better justify the average molar yield used in the model regardless to Riedel et al. (2016).

Line 335: The authors mentioned on Line 125 that the OPR was operated using an OH concentration of 10e+10 – 10e+13 molecule cm-3. At different places in the article, the authors refer to different ranges. This is confusing, thus it needs to be clarified.

Lines 343-347: As it is written in the article, the authors point out that the limitation of the reactive uptake of IEPOX is due to the acidity of the particles. Is it only due to the acidity, or the surface area, or both? Could the authors provide the surface area before and after injection of H2SO4 particles?

Lines 359-360: The statement, "oligomer decomposition could be fast in the ambient air", is not clear and a bit speculative. Could the authors provide more information and/or references to support this statement?

Lines 499-500 and general comment: Here the authors mentioned that they have estimated an OH yield based on range of OH concentrations of 10e+7 – 10e+10 molecule cm-3. However, in the experimental section, the range used was 10e+10 – 10e+13 molecule cm-3. Therefore, it is not clear if the yield was determined from extrapolation

in the model or from measurements. If it is from extrapolation, how reliable is the yield proposed in this study since different studies have shown that OH yield is dependent on the concentration of OH radicals?

In addition, the authors mentioned that they have investigated the impact of OH concentrations on OH yield. Regardless the previous studies, this set of experiments was conducted in a substantially small range to make a conclusion in the dependence of OH yields on OH concentration. Therefore, it appears to be overreached to propose such conclusions, and the question of the larger reactivity of IEPOX-derived SOA using a lower concentration of OH radicals remains present.

Technical comments:

Line 140: O3: subscript "3"

Line 141: s-1: superscript "-1"

Line 277: typo error. "isis" should be "is"

Lines 324/366: acidic NH4HSO4 should be either acidic (NH4)2SO4 or NH4HSO4

Line 418: > 10–14 should be 10-14

References: Arangio et al., 2015; J Phys Chem A; 119; 4533-4544

Budisulistiorini et al., 2013; ES&T; 47, 5686-5694

Budisulistiorini et al., 2015; ACP; 15; 8871-8888

Chen et al., 2015; ACP; 15; 3687-3701

Kroll et al., 2015; J Phys Chem A; 119; 10767-10783

Lai et al., 2015; PCCP; 15, 10953-10962

Lin et al., 2012; ES&T; 46; 250-258

Riedel et al., 2016; ACP; 16; 1245-1254

Riva et al., 2016; ES&T; 50; 5080-5088

Robinson et al., 2011; ACP; 11; 9605-9630

Xu et al., 2015; PNAS; 112, 37-42

---

## Referee Comment (RC2) · Anonymous Referee #2 · 5 Jul 2016

This paper describes detailed field measurements of "IEPOX-SOA", secondary organic aerosol deriving the reactive uptake IEPOX, a major product of isoprene oxidation. Key results include evidence that IEPOX-SOA is quite low in volatility, suggesting the importance of accretion products (consistent with other recently-published work), and measurements of the rate of atmospheric degradation of IEPOX-SOA material (which is a fundamentally new result). These are important results, and overall this is a solid study of general interest to the atmospheric chemistry community. It is certainly publishable in ACP; first, I have a number of detailed comments, listed below, that should be addressed prior to publication.

246-263 (and elsewhere): There manuscript makes repeated reference to the IEPOX-

SOA compounds being oligomers. However, it seems to me that other accretion products (namely organosulfates) could also explain much of the data (such as the low volatility of the compounds). The authors should either explain why they organosulfate formation is unlikely, or change the language throughout the paper to reflect this possibility.

Figs 1-2: A comparison with the sulfate thermogram would be helpful here for context.

273-276: These two possibilities seem closely related to me. The "real" volatility distribution (from physical volatilization only) is irrelevant if the room temperature evaporation is governed by chemical decomposition to more volatile monomers. The TD is measuring "effective volatility", which takes into account both physical evaporation and oligomer decomposition; this effective volatility (at atmospheric temperatures at least) is what matters for phase partitioning in the atmosphere. Therefore I'm not sure it's correct to say this approach is overestimating the volatility distribution.

279-280: The Vaden reference may be describing a very different effect, namely slow evaporation of monomers out of low-viscosity (and very dry) particles.

314-316: Here it is argued that wall loss of IEPOX cannot happen because of the high saturation vapor pressure of the molecule. But this is the wrong quantity to use, since IEPOX condenses almost entirely via reactive uptake. (If saturation vapor pressure is what determined IEPOX condensation, there'd be no such thing as IEPOX-SOA!) It is highly likely that there is reactive uptake to the walls, since there's probably a reasonable amount of sulfate (aqueous, likely quite acidic) from previous deposition. This effect needs to be included in these calculations.

351-354: This is an important definition, because it specifically excludes reactions that lead to the reaction of IEPOX-SOA components without major changes to the PMF factor (namely, with little change to m/z 82). This should be mentioned explicitly, as should the implication that these rates and uptake coefficients may be lower limit values.

378: "IEPOX-SOA" should be in brackets to signify concentrations (as written, the left side of formula looks like IEPOX minus SOA/IEPOX minus SOAo).

379-384: "ith OH exposure step" is an unusual (and to me, confusing) way to describe chemical kinetics. A better way to word this is in terms of the integrated OH exposure, up to some reaction time t.

411-412: the paper cited here (Slade and Knopf 2014) did not attribute the RH effect to differences in surface area, as stated. The effect described really derives from the surface-area-to-volume ratio, which is well known to have an influence on the kOH value; Robinson 2006 may be the more appropriate reference here. (Though it may have been derived even before then?)

480-484: This is a long, wordy way to argue that the V/SA ratio is equal to D/6, and that the assumption that the reacted species is well-mixed (with the mass fraction the same at the surface and in the bulk) simplifies the calculations. These have been inherent to all heterogeneous oxidation studies going back to at least Robinson et al (2006), so probably does not need to be included here.

494-496: Based on the paper cited, and the text immediately following (lines 497-498), the authors appear to be arguing that this implies a role of oxidative processing of the IEPOX within the aqueous phase. But organosulfate formation (or even the catalytic promotion of IEPOX uptake by sulfate) would also seem to be a reasonable explanation for this effect, with no additional oxidation required.

514-517: Probably more important than changes to gamma at higher [OH] is changes at lower (atmospherically relevant) [OH]. This effect has been observed previously [Che, et al. 2009, PCCP 11, 7885–7895].

518: this statement appears to be directly contradicted in line 546.

Figure 3: An extra pathway should be included, since the SOA "yield" from Riedel et al was used (lines 319-325). This small yield implies that most (∼90%) of the IEPOX

reactive uptake leads to non-SOA (i.e., gas-phase) products, different from those generated by OH reaction; the formation of these should be therefore included in the figure as well.

Figure 7: The caption (or legend) should explain what the different symbols mean.

———————————————

---

## Author Response (AR1)

**Response to reviewers for the paper "Volatility and lifetime against OH heterogeneous reaction of ambient Isoprene Epoxydiols-Derived Secondary Organic Aerosol (IEPOX-SOA)" by Weiwei Hu et al.**

We appreciate the reviewer's comments and support for publication of this manuscript after minor revisions. Following the reviewer's suggestions, we have carefully revised the manuscript. To facilitate the review process, we have copied the reviewer comments in black text. Our responses are in regular blue font. We have responded to all the referee comments and made alterations to our paper **(in bold text).**

**Anonymous Referee #1**

General Comments

R1.0. This is a very interesting study about the volatility and lifetime of IEPOX-derived SOA. As the authors point out in the manuscript, studies on heterogeneous reactions of ambient SOA with hydroxyl radicals are missing. This work presents a well-conduct series of ambient measurements and box models that provide supporting evidences for their findings. The study is technically well done and the paper is well written. I think it is suitable for publication in ACP after few minor revisions. My major comment is on the determination of the IEPOX-derived SOA lifetime as discussed below.

**A1.0:** We thank the reviewer for his/her review and useful comments. All of the items mentioned here are addressed in response to the more specific comments below.

**R1.1.** Line 70: The authors should consider adding the references and not only cite their previous work. Add these references: Robinson et al., 2011; Budisulistiorini et al., 2013; 2015; Chen et al., 2015, Xu et al., 2015

**A1.1:** The sentence in question states "IEPOX-SOA [...] can account for 6-34% of total OA over multiple forested areas across the globe range". Hu et al., 2016 was the paper that summarized all the results and showed that IEPOX-SOA is globally important and positively correlated with modeled gas-phase IEPOX. Thus we think the citation of this sentence is correct. The other references suggested by the reviewer are cited in different parts of our paper, as appropriate for each instance. There are also numerous citations in Hu et al., 2015, beyond those proposed by the reviewer here. For clarity we have revised that text to read:

**Line 68-70: "IEPOX-SOA measurements in field studies show that it can account for 6-34% of total OA over multiple forested areas across the globe, with important impacts on the global and regional OA budget (Hu et al., 2015 and references therein)."**

**R1.2.** Line 72: Add Riedel et al., 2015

**A1.2:** Added.

**R1.3.** Line 82: Similar comment as Line 70, add references: Robinson et al., 2011; Lin et al.,2012; Budisulistiorini et al., 2013

**A1.3:** Hu et al., 2015 is the only paper that (to our knowledge) has proposed the use of $f_{82}$ as a real-time tracer for estimating IEPOX-SOA mass concentration, which is what the sentence states. The other references suggested by the reviewer had already been cited above this sentence where they were relevant. Adding them to this sentence would be incorrect and confusing. Thus, this text has not been changed.

**R1.4. Line 85: Add reference: Kroll et al., 2015**

**A1.4:** We agree with reviewer's suggestion. Note that Kroll et al., 2015 has already been cited in the ACPD version of this study to support the volatilization of aerosol products. The updated text reads:

**"which also showed that OH oxidation led to formation of volatile products escaping to the gas phase (Kessler et al., 2010; Kroll et al., 2015)"**

**R1.5.** Line 114: How did temperature affect the reactivity of IEPOX-derived SOA? Indeed as shown by Lai et al. (2015), small changes in the temperature (20C-30C) could impact OH heterogeneous oxidation.

**A1.5:** In Lai et al. (2015) $k_{OH}$ increases with temperature. However, in SOAS $k_{OH}$ increases with RH (Figure 6), which we interpret as being mainly due to the increase in surface area due to water uptake. Ambient temperature is inversely correlated with RH, so the apparent trend in our study is that $k_{OH}$ *decreases* as temperature *increases*, which on the surface is the opposite effect than Lai et al. (2015) reported. There are many differences between the two studies (different species, lab vs field, particles on a sample holder vs suspended, etc.), and importantly it is not possible to isolate the effect of temperature from those of other variables in our ambient study. Thus we have added the following text recommending further study of this topic:

Line 464-467: **"An effect of temperature on $k_{OH}$ was not apparent in our study. Lai et al. (2015) reported a significant effect for a laboratory study with a pure compound. We recommend that this issue is explored further in the laboratory using pure IEPOX-SOA."**

**R1.6.** Line 125: Why did the authors limit the range of OH exposures from 10e+10 – 10e+13 molecule cm-3? Indeed, as previously reported and discussed below, OH uptake and heterogeneous oxidation is impacted by the concentration of OH radicals (Arangio et al., 2015).

**A1.6:** First of all, we have to clarify that the units for OH exposure are molec. cm$^{-3}$ s, not molec cm$^{-3}$. The range of OH exposure of $10^{10}$-$10^{13}$ molecule cm$^{-3}$ s corresponds to 0.1 day to several months equivalent aging time, which is the range over which it is important to investigate atmospheric aerosol chemistry. We modified the main text for clarity to read:

Line135-143: **"A large range of OH exposures ($10^{10}$-$10^{13}$ molec. cm$^{-3}$ s) can be achieved by varying UV light intensity, equivalent to several hours to several weeks of photochemical aging of ambient air (assuming a 24-hr average OH=$1.5\times10^{6}$ molec. cm$^{-3}$; Mao et al., 2009). Thus we believe that the range of OH exposures ($10^{10}$-$10^{13}$ molec. cm$^{-3}$ s) covered by our study is the relevant range for the atmosphere. We note that OH radical concentration can be calculated as the ratio of the OH exposure ($10^{10}$-$10^{13}$ molec. cm$^{-3}$ s) and the residence time (200 s). The calculated OH radical concentration in our flow reactor is between $5\times10^{7}$**

**to $5 \times 10^{10}$ molec. cm$^{-3}$. The lower range of OH radical concentration is comparable to the higher end of observed ambient OH concentrations (Mao et al., 2009)."**

**R1.7.** Line 154: The authors mentioned other type of oxidants, such as O3 and NO3. If they performed some experiments using different oxidants, it would be interesting to discuss the reactivity of these oxidants (especially NO3) and compare the results with OH radicals. Could the authors provide additional experimental information for the production of NO3 radicals in the OFR?

**A1.7:** We indeed conducted ambient OFR experiments with O$_3$ and NO$_3$, in addition to OH during the SOAS study. However, the results from those experiments greatly exceed the scope of this paper, and will be presented in future publications. For example, we are close to submitting a stand-alone paper that reports results of OFR experiments with O$_3$ and NO$_3$ for a different field study, and that paper has over 40 pages of text and 10 figures.

**R1.8.** Line 155: The authors mentioned that they used different methods (i.e. OFR 185 and OFR 254?) but decided to present only results from one. Could they explain the reasons and add some information and/or comparison between both methods?

**A1.8:** This is incorrect. All experiments reported here were done with the OFR185 method, as clearly stated in Line 155 of the ACPD paper. In fact, the words "OFR 254" do not appear in our paper at all. We refer the reviewer to the work of Palm et al. (ACP 2016) who presented a comparison of results from the OFR185 and OFR254 methods for the BEACHON-RoMBAS study. Our conclusion from that and other studies is that although the results are similar, the OFR185 method is preferable, especially for ambient studies, and thus that was the method used during SOAS. The main text has been modified to reflect this:

**Line 162-165: "A comparison of results from the OFR185 and OFR254 methods for a study at a pine forest was presented by Palm et al. (2016), showing similar SOA formation by both methods. Their results, together with other model studies (Peng et al., 2015) showed that the OFR185 method is preferable for ambient studies, and thus that was the method used during SOAS."**

**R1.9.** Line 168-170: The authors mentioned that during the dry season SOA were significantly influenced by biomass burning. This statement is supported by neither references nor experimental data. Additional information is needed to evaluate the importance of BBOA in OA. In addition, how do the authors expect the presence of BBOA to impact the IEPOX-SOA aging?

**A1.9:** The presence of BBOA was very obvious during the dry season. We have added a reference to Martin et al. (ACP 2016) which discusses this issue for the GoAmazon study.

We have added the following text in Line 435-438 to address the second point:

**"The higher aerosol concentrations from biomass burning during the Amazon study did not appear to cause any major differences in the observed OH uptake. This may be due to the mostly liquid state of the ambient particles in both studies, which will be discussed in detail below (Bateman et al., 2015; Pajunoja et al., 2016)."**

**R1.10.** Line 192: Similar comment as line 82, add the references.

**A1.10:** Several references have been added.

**"(Robinson et al., 2011; Lin et al., 2012; Allan et al., 2014; Hu et al., 2015)".**

**R1.11.** Lines 203-204: Riedel et al. (2016) have recently reported bulk laboratory measurements and determined the reaction rate constants of IEPOX-SOA products from the reactive uptake of IEPOX. The authors didn't consider this study and only use a global rate constant. The rates constants proposed by Riedel et al. (2016) should be added in the model and results from both approaches should be compared in the paper.

**A1.11:** In our paper we use the model of the uptake of gas-phase IEPOX to form IEPOX-SOA to show that it is negligible in the reactor, and thus that our measurements can be directly interpreted as the heterogeneous oxidation of IEPOX-SOA in the aerosol. The conclusion that IEPOX-SOA formation is negligible is strengthened if we consider the timescale of IEPOX-SOA formation in Riedel et al (2016). We have added the following text to the paper to clarify this issue:

Line 386-389: **"If the more detailed IEPOX-SOA formation model of Riedel et al. (2016) were used, the modeled IEPOX-SOA formation would be significantly lower, due to the consideration of the kinetics of IEPOX-SOA formation. That reinforces our conclusion that IEPOX-SOA formation in the reactor was negligible."**

**R1.12.** Lines 270-271: At which temperature did the authors perform these evaporation experiments?

**A1.12:** We did not perform evaporation experiments in this part of the study, but estimated the isothermal evaporation loss upon dilution, based on the volatility distribution estimated from the thermodenuder (TD) measurements, using Eq 1.

To clarify, the original sentence "Using the volatility distributions determined from the TD, the fractional losses for both OA and IEPOX-SOA due to evaporation upon dilution can be estimated" was revised to read:

Line 279-281: **"It is of high interest to estimate the fractional losses for both OA and IEPOX-SOA due to isothermal evaporation upon dilution. These losses can be estimated using the volatility distributions estimated from the TD measurements."**

**R1.13.** Lines 295-296: Riva et al. (2016) have shown that IEPOX reactive uptake could be significantly reduced by the presence of organic on seed aerosols. The authors should add this reference and also discuss this potential impact in their model.

**A1.13:** The result of reduced IEPOX uptake due to organic coatings was already reported by Gaston et al. (2014), and it was incorporated in their model, which we use here. We also show (Fig. S16) that even if the reduced uptake is neglected, IEPOX-SOA formation is still negligible. Therefore, we have not changed the paper in response to this comment, other than to add a citation to Riva et al. (2016) when discussing the slower uptake due to organic coatings in Line 314.

**R1.14.** Lines 321-323: The authors should better justify the average molar yield used in the model regardless to Riedel et al. (2016).

**A1.14:** We are not totally clear of what the reviewer means by "regardless to Riedel et al." The molar yield used is supported from a literature reference of an experimental study. See response R1.11 for further details on this issue.

**R1.15.** Line 335: The authors mentioned on Line 125 that the OPR was operated using an OH concentration of 10e+10 – 10e+13 molecule cm-3. At different places in the article, the authors refer to different ranges. This is confusing, thus it needs to be clarified.

**A1.15:** The reviewer appears to confuse OH concentrations (molec $cm^{-3}$) with OH exposures (integral of OH x time, units of molec. $cm^{-3}$ s). Both quantities are related, and are relevant for different reasons. See response to comment A1.6 for further details and text modified for clarity.

**R1.16.** Lines 343-347: As it is written in the article, the authors point out that the limitation of the reactive uptake of IEPOX is due to the acidity of the particles. Is it only due to the acidity, or the surface area, or both? Could the authors provide the surface area before and after injection of H2SO4 particles?

**A1.16:** The increased surface areas indeed played a role for the enhanced IEPOX-SOA formation in the OFR, however its role is minor compared to that of the acidity changing. Following the reviewer's suggestion, we have modified the main text to read:

Line 374-378: **"The increased surface area and acidity from added $H_2SO_4$ seed both help accelerate IEPOX reactive uptake, although acidity plays a more important role. If we use in the model the ambient surface area and pure $H_2SO_4$ the lifetime of IEPOX uptake is ~10 min, while if we assume the ambient acidity and the same surface area as 100 µg $m^{-3}$ of pure $H_2SO_4$, the lifetime is 1.1 h."**

**R1.17.** Lines 359-360: The statement, "oligomer decomposition could be fast in the ambient air", is not clear and a bit speculative. Could the authors provide more information and/or references to support this statement?

**A1.17:** This is indeed speculative, as this possible process has not been reported in the literature. The main idea here is to emphasize that even if fast decomposition of IEPOX-SOA oligomers happened in the ambient air, the decomposition would not have played an important role for IEPOX-SOA loss in the flow reactor because the residence time of aerosols in the OFR is low (3 min). Thus, to clarify, the text has been revised to read:

Line 392-397: **"Oligomer decomposition followed by evaporation is very likely negligible in the flow reactor residence time scale of 3 min. However, this process could be more important in ambient air, and if fast, could influence the IEPOX-SOA lifetime. No results for oligomer decomposition rates or extents for IEPOX-SOA have been reported in the literature, to our knowledge. Thus, further research on this topic is recommended."**

**R1.18.** Lines 499-500 and general comment: Here the authors mentioned that they have estimated an OH yield based on range of OH concentrations of 10e+7 – 10e+10 molecule cm-3. However, in the experimental section, the range used was 10e+10 – 10e+13 molecule cm-3. Therefore, it is not clear if the yield was determined from extrapolation in the model or from measurements. If it is from extrapolation, how reliable is the yield proposed in this study since different studies have shown that OH yield is dependent on the concentration of OH radicals?

**A1.18:** Again this appears to be a confusion between OH concentrations and exposures, see response A1.6.

**R1.19.** In addition, the authors mentioned that they have investigated the impact of OH concentrations on OH yield. Regardless the previous studies, this set of experiments was conducted in a substantially small range to make a conclusion in the dependence of OH yields on OH concentration. Therefore, it appears to be overreached to propose such conclusions, and the question of the larger reactivity of IEPOX-derived SOA using a lower concentration of OH radicals remains present.

**A1.19:** Our OFR study used a OH concentration range of 1000, which is similar to that used in other numerous laboratory studies listed in Table 1. E.g., Che et al., 2009; Smith et al., 2009; Kessler et al., 2010; Kessler et al., 2012; Slade and Knopf, 2014. The lower range of OH radical concentration in our study is also comparable to the higher end of observed ambient OH concentrations (Mao et al., 2009), see revised text in A1.6. However, it is important to note that our results for $\gamma_{OH}$ do not cover the range of atmospheric OH levels. We have added the following text to the paper to clarify this point:

Line 558-560: **"We note that our experiments do not rule out some dependence of $\gamma_{OH}$ on OH at lower OH levels in the atmosphere."**

**Technical comments:**

**R1.20.** Line 140: O3: subscript "3"

**A1.20:** Corrected

**R1.21.** Line 141: s-1: superscript "-1"

**A1.21:** Corrected

**R1.22.** Line 277: typo error. "isis" should be "is"

**A1.22:** Corrected

**R1.23.** Lines 324/366: acidic NH4HSO4 should be either acidic (NH4)2SO4 or NH4HSO4

**A1.23:** Revised to be "acidic $(NH_4)_2SO_4$" and "acidified $(NH_4)_2SO_4$ seed"

**R1.24.** Line 418: > 10–14 should be 10-14

**A1.24:** Corrected

**Anonymous Referee #2**

**General Comments**

**R2.0.** This paper describes detailed field measurements of "IEPOX-SOA", secondary organic aerosol deriving the reactive uptake IEPOX, a major product of isoprene oxidation. Key results include evidence that IEPOX-SOA is quite low in volatility, suggesting the importance of accretion products (consistent with other recently-published work), and measurements of the rate of atmospheric degradation of IEPOX-SOA material (which is a fundamentally new result). These are important results, and overall this is a solid study of general interest to the atmospheric chemistry community. It is certainly publishable in ACP; first, I have a number of detailed comments, listed below, that should be addressed prior to publication.

**A2.0:** We thank the reviewer for his/her review and useful comments. All of the items mentioned here are addressed in response to the more specific comments below.

**R2.1.** 246-263 (and elsewhere): There manuscript makes repeated reference to the IEPOX-SOA compounds being oligomers. However, it seems to me that other accretion products (namely organosulfates) could also explain much of the data (such as the low volatility of the compounds). The authors should either explain why they organosulfate formation is unlikely, or change the language throughout the paper to reflect this possibility.

**A2.1:** Here, we include the IEPOX organosulfate as part of IEPOX-SOA. As reported in the ACPD version of this paper (L234-236), the IEPOX organosulfate accounts for 24% of total IEPOX-SOA. We have added the following text to the manuscript to address this point:

Line 260-266**: "Although the IEPOX organosulfate may have lower volatility than estimated in Fig. 1D, it only accounts for 24% of total IEPOX-SOA (Hu et al., 2015) and thus it cannot be the only reason for the low volatility of the bulk of IEPOX-SOA. For reference, only 5% of the total sulfate is due to the IEPOX organosulfate, with the rest being inorganic sulfate consistent with other results from the SE US in Summer 2013 (Liao et al., 2015). Indeed, the thermogram of total sulfate is very different from that of IEPOX-SOA (Fig. 1a and Fig.2a)."**

**R2.2.** Figs 1-2: A comparison with the sulfate thermogram would be helpful here for context.

**A2.2:** As discussed in A2.1, only 5% of the total sulfate is due to the IEPOX organosulfate, and indeed the two thermograms are quite different. This has been added to Figure 1A and Figure 2A as shown below.

[Figure]

**Figure 1** (a) Mean mass fraction remaining of IEPOX-SOA, OA and SO₄ versus temperature in TD ("thermograms") during SE US study. (b) Volatility distributions of IEPOX-SOA and OA estimated from TD thermograms (see text). Bars are offset for clarity and were both calculated for integer log(*C**) values. (c) Evaporation losses of IEPOX-SOA and OA as a function of dilution factors. (d) Volatility of typical IEPOX-SOA molecular species in the aerosol phase based the on SIMPOL group contribution method (Pankow and Asher, 2008). The reduction in vapor pressure upon addition of a nitrate group was used to estimate the effect of the sulfate group, due to lack of SIMPOL parameters for the latter, and the derived *C** may be overestimated for this reason.

[Figure]

**Figure 2** (a) Thermogram of OA, $SO_4$ and background-corrected $C_5H_6O^+$ ion in the SE US and Amazon studies. (b) Volatility distributions of $C_5H_6O^+$ and OA estimated based on TD thermograms from the Amazon study.

**R2.3.** 273-276: These two possibilities seem closely related to me. The "real" volatility distribution (from physical volatilization only) is irrelevant if the room temperature evaporation is governed by chemical decomposition to more volatile monomers. The TD is measuring "effective volatility", which takes into account both physical evaporation and oligomer decomposition; this effective volatility (at atmospheric temperatures at least) is what matters for phase partitioning in the atmosphere. Therefore I'm not sure it's correct to say this approach is overestimating the volatility distribution.

**A2.3: W**e respectfully disagree. The volatility distribution of the molecules present in the aerosol at any given time depends on their molecular identity. That distribution may indeed be even lower than estimated in Fig. 1b, if some of the oligomers are decomposing in the thermal denuder, and thus evaporating at lower temperatures than would be needed to evaporate the intact oligomer. Lopez-Hilfiker et al. (2016) has shown molecular evidence that this process is indeed occurring with a related thermal desorption mass spectrometric instrument during SOAS. Thus we do believe that the sentence starting with **"One is that…"** is indeed correct.

The effect of oligomer decomposition under ambient conditions would be more complex, when the airmass is undergoing dilution. As the gas-phase is depleted of monomers by dilution, some of the molecules comprising the oligomers would not return to the particle phase after evaporation. However, the volatility distribution of the molecules present in the particle phase would still be similar to that measured in SOAS (assuming all oligomers have consistent rates).

Thus we believe that text **("One is that the real volatility distribution of IEPOX-SOA is likely even lower, since the TD results are thought to be affected by oligomer decomposition upon heating. The other one is that this calculation neglects the effect of possible decomposition of oligomers into monomers in ambient air.")** is correct, and that both possibilities are distinct and both need to be considered, and have made no changes in response to this comment.

**R2.4.** 279-280: The Vaden reference may be describing a very different effect, namely slow evaporation of monomers out of low-viscosity (and very dry) particles.

**A2.4:** Our understanding of the α-pinene SOA debates is that oligomers are thought to comprise a large fraction of the particle mass. However, the possibility suggested by the reviewer is also plausible. We have modified this sentence to read as:

Line 296-298: **"E.g. Vaden et al (2011) reported that it took 24 h to evaporate 75% of α-pinene SOA (although it is possible that processes other than oligomer decomposition were important for determining the timescale of those experiments)."**

**R2.5.** 314-316: Here it is argued that wall loss of IEPOX cannot happen because of the high saturation vapor pressure of the molecule. But this is the wrong quantity to use, since IEPOX condenses almost entirely via reactive uptake. (If saturation vapor pressure is what determined IEPOX condensation, there'd be no such thing as IEPOX-SOA!) It is highly likely that there is reactive uptake to the walls, since there's probably a reasonable amount of sulfate (aqueous, likely quite acidic) from previous deposition. This effect needs to be included in these calculations.

**A2.5:** We still believe that IEPOX loss to the walls of the OFR is negligible. We have added the following text to the paper to explain this point further:

Line 351-355: **"We can estimate the timescale of IEPOX loss rate to the walls by assuming that the walls are covered by a layer of deposited ambient aerosol. We combine the 1st order rate of collision of gas molecules with the walls (400 s; Palm et al., 2016) and the uptake coefficient for IEPOX-SOA in ambient aerosols ($\gamma_{IEPOX} = 0.009$) to estimate a timescale of IEPOX loss to the walls of 12.3 h, which is negligible compared to residence time of IEPOX (~200s) in the OFR. Even if the walls were covered by sulfuric acid ($\gamma_{IEPOX} = 0.082$), the timescale of loss would be 1.4 h."**

**R2.6.** 351-354: This is an important definition, because it specifically excludes reactions that lead to the reaction of IEPOX-SOA components without major changes to the PMF factor (namely, with little change to m/z 82). This should be mentioned explicitly, as should the implication that these rates and uptake coefficients may be lower limit values.

**A2.6:** We have added the following text to clarify the issues raised in this point:

Line 386-389: **"Note that the rate derived here may be a lower limit for individual molecular components of IEPOX-SOA, if e.g. it takes two or more OH reactions for their AMS spectrum to no longer resemble that of IEPOX-SOA."**

**R2.7. 378:** "IEPOX-SOA" should be in brackets to signify concentrations (as written, the left side of formula looks like IEPOX minus SOA/IEPOX minus SOAo).

**A2.7:** Revised as suggested.

**R2.8.** 379-384: "ith OH exposure step" is an unusual (and to me, confusing) way to describe chemical kinetics. A better way to word this is in terms of the integrated OH exposure, up to some reaction time t.

**A2.8:** We prefer to keep this as is, because in our experiments we change $OH_{exp,i} = [OH]_i \times \Delta t$ by changing $[OH]_i$ while keeping $\Delta t$ constant. However, we have simplified the expression by replacing $\Delta t_i$ by $\Delta t$, since that parameter is constant and does not change.

**R2.9.** 411-412: the paper cited here (Slade and Knopf 2014) did not attribute the RH effect to differences in surface area, as stated. The effect described really derives from the surface-area-to-volume ratio, which is well known to have an influence on the kOH value; Robinson 2006 may be the more appropriate reference here. (Though it may have been derived even before then?)

**A2.9:** We are unsure about which Robinson 2006 paper the reviewer is referring to, and a full citation is not provided. In any case the effect described by the reviewer is not what we believe is going on in our experiments. For particles of constant composition, indeed a higher surface/volume ratio will increase $k_{OH}$. However, as the ambient particles take up water, the surface/volume ratio *decreases*. We believe that the rate goes up because the added material (water) does not react with OH. Thus the surface area increase leads to more OH being taken up into the particle, but the mass of organic species that can react with that OH does not change. We found two papers from Allen Robinson et al. from 2006, but neither of them discusses the effect of water uptake on heterogeneous $k_{OH}$.

However, it is correct that Slade and Knopf, who worked in a different regime at much lower RH, attributed the effect to a different physical mechanism (high viscosity) which is not relevant for our studies, and thus we have removed that reference.

We are not aware of any previous paper reporting an RH effect for the reasons we stated in the paper, and thus that sentence does not have any literature reference in the revised version of our paper.

**R2.10.** 480-484: This is a long, wordy way to argue that the V/SA ratio is equal to D/6, and that the assumption that the reacted species is well-mixed (with the mass fraction the same at the surface and in the bulk) simplifies the calculations. These have been inherent to all heterogeneous oxidation studies going back to at least Robinson et al (2006), so probably does not need to be included here.

**A2.10:** We have simplified this text to read:

**"We assume IEPOX-SOA is uniformly mixed with the other aerosol species (both in the surface and volume), and independent of particle size. Then for a spherical particle $V_{total}$ /$S_{total}$ is equal to $d_{surf}$/6, where $d_{surf}$ is defined as surface-weighted particle diameter."**

We cannot find this stated in the Robinson papers we know of, so we have not cited that paper for this point.

**R2.11.** 494-496: Based on the paper cited, and the text immediately following (lines 497-498), the authors appear to be arguing that this implies a role of oxidative processing of the IEPOX within the aqueous phase. But organosulfate formation (or even the catalytic promotion of IEPOX uptake by sulfate) would also seem to be a reasonable explanation for this effect, with no additional oxidation required.

**A2.11:** The text in those lines refers to the observed carbon oxidation state ($OS_C$) of IEPOX-SOA in different situations. Indeed, ambient IEPOX-SOA has a higher average $OS_C$ than that freshly formed in chambers. Organosulfate formation does not affect $OS_C$, and thus it is not a plausible explanation for this effect. Uptake of IEPOX at a faster rate would form more fresh IEPOX-SOA, but not change its $OS_C$. Thus we have not changed the text in response to this comment.

**R2.12.** 514-517: Probably more important than changes to gamma at higher [OH] is changes at lower (atmospherically relevant) [OH]. This effect has been observed previously [Che, et al. 2009, PCCP 11, 7885–7895].

**A2.12:** The values of $\gamma_{OH}$ reported by Che et al. only varied by 10% over the range [OH] = 1-7 x $10^8$ molec. cm$^{-3}$, without a clear trend (their Table 1). Thus this variation is likely due to experimental uncertainty in those experiments. In the previous response (A1.19) we clarified that we did not cover the ambient OH concentration range in our experiments. We have added the following text to the paper to further clarify this point:

Line 558-560: **"We note that our experiments do not rule out some dependence of $\gamma_{OH}$ on OH at lower OH levels in the atmosphere. However, Che et al. (2009) found no effect of OH on $\gamma_{OH}$ for squalane particles in the range 1-7 x $10^8$ molec. cm$^{-3}$."**

**R2.13.** 518: this statement appears to be directly contradicted in line 546.

**A2.13:** There is no contradiction, but to avoid confusion we have clarified the text in Line 556-558 to read:

**"A possible explanation is that in our study OH uptake occurs on liquid particles, resulting on fast OH diffusion into the particle bulk, and causing OH uptake not to be limited by surface adsorption."**

**R2.14.** Figure 3: An extra pathway should be included, since the SOA "yield" from Riedel et al was used (lines 319-325). This small yield implies that most (_90%) of the IEPOX reactive uptake leads to non-SOA (i.e., gas-phase) products, different from those generated by OH reaction; the formation of these should be therefore included in the figure as well.

**A2.14:** This is a good suggestion. Figure 3 has been revised to address this point, and it is reproduced below:

[Figure]

**Figure 3** Mechanism diagram of gas-phase IEPOX model in ambient and OFR conditions. ISOPOOH-SOA refers to SOA formed through gas-particle partitioning of low-volatile VOCs from oxidation of isoprene 4-hydroxy-3-hydroperoxide (4,3-ISOPOOH) under low-NO conditions (Krechmer et al., 2015).

**R2.15.** Figure 7: The caption (or legend) should explain what the different symbols mean.

**A2.15:** This information was already included by using the axis colors. However, for further clarity we have also added a legend to Figure 7. Please see the revised Figure 7 below:

[revised manuscript text omitted]